

# Resolving the contributions of local emissions to measured concentrations: a method comparison

Taylor D. Edwards[1], Yee Ka Wong[1], Cheol-Jeon Heong[1], Jonathan M. Wang[1], Yushan Su[2], and Greg J. Evans[1]

[1]Department of Chemical Engineering and Applied Chemistry, University of Toronto, Wallberg Memorial Building, 184 College St., Toronto, Ontario, Canada
[2]Environmental Monitoring and Reporting Branch, Ontario Ministry of Environment, Conservation and Parks, 125 Resources Road, Toronto, Ontario, Canada.

*Correspondence to*: Greg J. Evans (greg.evans@utoronto.ca)

**Abstract.** To accurately study the characteristics of an air pollution emitter, it is necessary to isolate the contribution of that emitter to total measured pollution concentrations. A variety of published methods exist to complete this task, like placing measurements upwind the emitter, employing a distant background measurement station, or algorithmic methods that extract a background from the time-series of measured concentrations (e.g., wavelet decomposition). In this study, we measured nitrogen oxides ($NO_X$), carbon monoxide (CO), carbon dioxide ($CO_2$), and fine particulate matter ($PM_{2.5}$) at four sites spanning Toronto, Ontario, Canada. We first characterized the spatial variability of background concentrations across the city, and then tested the accuracy of seven different algorithmic methods of estimating true measured upwind-of-emitter backgrounds near Toronto's Highway 401 by using the data collected at a downwind site. These methods included time-series and regression methods, including machine learning (XGBoost). We observed background concentrations had notable spatial variability, except for $PM_{2.5}$. When predicting backgrounds upwind the highway, we found a distant measurement station provided an accurate background only during some times of day and was least accurate during rush hours. When testing algorithmic predictions of upwind-of-highway backgrounds, we found that regression models outperformed time-series methods, with best predictions having $R^2$ exceeding 0.75 for all four pollutants. Despite the better performance of regression models, time-series methods still provided reasonable estimates; we also found that emitter-specific covariates (e.g. traffic counts, onsite dispersion modelling) did not play an important role in regressions, suggesting backgrounds can be well-characterized by time of day, meteorology, and distant measurement stations. Based on our results, we provide ranked recommendations for choosing background estimation methods. We suggest future air pollution research characterizing individual emitters include careful consideration of how background concentrations are estimated.

## 1    Introduction

Across air pollution literature, there is a common distinction between stationary field measurement sites located well away from any known sources that record *background* pollution concentrations and those that record *local* concentrations, such as



near-road sites, influenced by emissions from nearby "local" sources. Generally, background concentrations are considered to arise from a mix of more distant upwind anthropogenic and natural sources and processes, while local concentrations are impacted by one or more nearby sources of interest. The difference between the concentration of an air pollutant measured at near-source and background sites can be attributed to local emissions. Within this process of apportioning the measured total
concentration, the contribution of emissions from nearby sources is referred to as the *local* or *emitted* concentration, while that within air masses arriving from upwind of a measurement site is referred to as the *background* concentration.

Good measures of background concentrations are important for isolating local sources of pollution. Ideal outdoor field measurements would include instruments both up- and down-wind of the source of interest, such that the source's contribution is the difference between the two. However, this is not always possible: requiring two simultaneous measurements increases
instrumentation and operation cost; there may not be an appropriate upwind location to place instruments; and widely varying predominant wind directions might necessitate more than just one upwind-downwind measurement pair. For these reasons, tools for estimating background concentrations ($C_{bkg}$) without a second measurement site are valuable. With reliable $C_{bkg}$ estimates, researchers can isolate continuous measurements of their sources of interest, which is vital for source attribution and measuring emission rates and emission factors.

Traditionally, if measurements immediately upwind of a source of interest were not available, researchers might utilize either an urban background station or tracer species to isolate contributions from sources of interest. Urban background stations are typically within a few kilometres of the study location but are removed from any major nearby sources; and these might be located in a park or a nearby rural area. Tracer species are those that are specific to the source of interest – if a researcher knows a measured emissions source is the only major nearby source of a particular species, they can be confident their
measured source is the only contributor to measured concentrations of that species.

Unfortunately, both approaches, despite their prevalence in the literature, have limitations. Urban background stations might not be wholly isolated from all sources or background concentrations might vary spatially between the urban background station and the study site (particularly in the context of the strict definition of "background concentration" we provide below). For tracer species, in many cases the source of interest cannot be guaranteed to be the only measured contributor. For example,
$NO_x$ is often considered a tracer for traffic emissions, but in a dense urban area measured nitrogen oxides ($NO_x$) concentrations will contain emissions from many different roads, so no single traffic type or road can be isolated.

Beyond these common approaches, there exist some other methods for estimating background concentrations, particularly for application to continuous time-series measurements of atmospheric pollution: examples are signal decomposition and estimating dilution between source and receptor. In summary, notable background-subtraction methods include:
• Measuring pollutant concentrations immediately upwind of the source of interest, as mentioned above, such as in highway studies by Zhu et al. (2002), Kohler et al. (2005), and Frey et al. (2022).
   • Designating a geographically distinct measurement station as an urban or regional background, with that station typically having few nearby emissions sources (Hicks et al., 2021; Hilker et al., 2019).
   • Comparing times when a measurement site is up- and down-wind of a target source (Hilker et al., 2019).



- Identifying a background or apportioning sources via wavelet decomposition (Klems et al., 2010; Sabaliauskas et al., 2014; Wei et al., 2019).

- Wang (2018) and Hilker et al. (2019) employed and tested an iterative algorithm that heuristically estimates a background signal similar to that produced from wavelet decomposition, which is termed as *pseudo-wavelet*. In brief, this method takes a smoothed interpolation of minima in the measured near-source concentration within a moving time window.

- Inverse dispersion modelling, where multiple downwind measurements are paired with a dispersion model estimating downwind concentrations given an emission rate. This approach requires multiple geographically distinct measurements downwind of the source of interest. Inverse dispersion modelling approaches are usually applied to measure emission rates from the source of interest, though concentration upwind of the emitter should be produced as a by-product of this calculation (Fushimi et al., 1997; Olaguer, 2022).

- Clustering algorithms: clustering can identify sources by grouping correlated pollutants, and may not necessarily delineate between local and background sources, however Rodríguez et al. (2024) demonstrated a separation of local and non-local sources using a fuzzy clustering algorithm.

- Geospatial interpolation from urban background stations, which can estimate the spatial variability of background concentrations, such as in Arunachalam et al. (2014).

- Localized iterative regression within a time-series of concentrations to extract a baseline signal, as described by Ruckstuhl et al. (2012); however this study presented a method to further decompose measurements from a background site, implying a definition of background concentration that is geographically broader than what we consider in this study.

## 1.1    Defining "background concentration"

To address the limitations of urban background sites and the other methods identified above, we propose a definition for background subtraction that is useful for isolating emissions sources of interest: background concentrations, $C_{bkg}$, are only the portions of the total measured concentrations that were not emitted from the local emission source of interest. This definition is similar to the one provided by Arunachalam et al. (2014). With this definition, the total measured concentration, $C_{meas}$, is strictly a sum of the local concentration, $C_{local}$, and background concentration, $C_{bkg}$:

$$C_{meas} = C_{local} + C_{bkg} \tag{1}$$

As a corollary to this definition, $C_{local}$ is only the portion of $C_{meas}$ that was emitted from the source of interest, and thus the local concentration becomes useful for estimating emissions, source characteristics, etc. This definition recognizes that the background concentration may vary across regions such as a city because of the many sources present. At the same time, the background concentration across a city can be relatively homogenous, if much of the background originates from sources or





processes well upwind of a city, as is often the case for pollutants such as $PM_{2.5}$ and $CO_2$. Ideally, this background concentration should be measured directly upwind the source of interest, with no interstitial sources. The up- and down-wind measurements should also be near enough to each other and the emissions source that dilution of background concentrations while they travel

between the up- and down-wind instruments is not of concern. Total measured concentration, $C_{meas}$, is then the concentration downwind the source. This is the configuration at the highway field site studied here, which had instruments placed up- and down-wind a major urban highway in Toronto, Canada. While it is desirable for the background site to be as close as possible to the emissions source of interest, such as directly upwind a busy road, the nearer the background site is to the emission source, the greater the potential for emissions from that source to contribute at times to the concentrations measured at the

background site. We posit that this definition of background concentration lends itself readily to useful measurements of $C_{local}$. Accordingly, it is desirable that researchers measuring rates and/or characteristics of emissions sources can estimate $C_{bkg}$ when direct measurement is not possible, as previously discussed.

    We note that this definition differs from existing interpretations of *background* in air pollution research, where background might be interpreted as either a minimum or baseline concentration, or as pollution arising from long-range transport from

multiple distant sources (Gómez-Losada et al., 2016, 2018). These existing definitions would imply homogeneous and temporally constant concentrations spread across an entire neighbourhood, city, or region. Measuring such a background concentration might require rural measurement, or an urban measurement isolated from any single source. In our case, we are interested in measuring $C_{bkg}$ for the purpose of extracting $C_{local}$, so emissions from sources other than the targeted emitter are only a problem if they are so nearby as to render the measurement of $C_{bkg}$ obviously unusable.

**1.2**     **Study outline and objectives**

    In this study we tested, qualitatively and quantitatively, the accuracy of a variety of methods for estimating background concentration at a field site next to a large roadway emissions source. We first qualitatively examined the extent to which background concentrations varied across an urban area, and tested how accurately concentrations measured at two distant urban background stations matched background concentrations measured at the highway site. We then tested the accuracy of

seven algorithms for predicting background concentrations at the near-road site. The algorithmic methods were differentiated into two classes: *frequency methods* used the time-series nature of $C_{meas}$ to predicted $C_{bkg}$, on the theoretical basis that background concentrations vary on a longer temporal scale than a nearby source, and that $C_{bkg} = C_{meas}$ at least occasionally. Frequency methods included the pseudo-wavelet method presented by Wang (2018) and Hilker et al. (2019). *Regression methods* were those that incorporated additional covariates measured or estimated at the study site and were regressed to the

measured upwind background concentrations. Regression methods included both traditional linear regressions like ordinary least squares (OLS) and machine learning models like XGBoost (Chen and Guestrin, 2016).



The objective of this study was to improve our understanding of how background concentrations vary across an urban area, and to evaluate if measured $C_{bkg}$ can be reliably estimated from $C_{meas}$ and other covariates at a downwind site. Specifically, our objectives, questions, and relevant hypotheses were:

- Do background concentrations vary geographically across an urban area, and are there times or conditions where background concentrations are homogeneous? We hypothesized that background concentrations will differ geographically across the city, and that this inter-site difference will be greater for pollutants with more geographically variable emissions like $NO_x$ and carbon monoxide (CO).

- Evaluate the ability of a variety of algorithmic methods for estimating background concentrations from measured
concentrations downwind of a source of interest. For the site studied here, the source of interest was a major urban highway. We hypothesized that regression methods will outperform frequency methods, on the basis that the additional information provided by the extra variables will improve estimates.

- Evaluate qualitatively and quantitatively the usefulness of the tested background concentration estimates for arbitrary new measurements and make a recommendation for method(s) to algorithmically estimate background
concentration for future urban air pollution studies.

- Test the extent to which low-cost air quality sensors can resolve the contributions of local emissions and background concentrations at a near-road site.

In all cases we optimized and assessed the accuracy of $C_{bkg}$ predictions chiefly via the root mean square error (RMSE). We also present and consider other quantitative metrics of prediction performance and examine qualitative performance from
various figures of true and estimated $C_{bkg}$. We evaluate each of these objectives, questions, and hypotheses in our results below. We validated the accuracy of each algorithmic estimates of background concentration by temporarily deploying a low-cost air pollution sensor platform to the upwind side of the tested highway site – this approach of short-term low-cost deployments is becoming increasingly feasible with the growing availability of competitive low-cost sensor products. This study thus also serves as an example of the benefits of such sensor products.

This study was completed as part of the larger Study of Winter Air Pollution in Toronto (SWAPIT) campaign, a collaborative effort between the academic, government, and private institutions in the Toronto, Ontario region.

## 2    Methodology

### 2.1    Field measurements

We gathered field measurements at five sites throughout Toronto, Ontario, Canada, from 2023-11-23 to 2024-04-12,
totalling just over 141 days of measurements. The next two sections describe the sampling sites and instruments.





### 2.1.1 Site descriptions

The primary highway field site was located adjacent a stretch of Toronto's Highway 401 located at approximately UTM 617300 m E 4840900 m N 17T. This stretch of highway is one of the busiest in North America, with over 400,000 annual average daily traffic (AADT) in 2016 as reported by the Ontario Ministry of Transportation (2016). It is 17 lanes and 113 m
wide adjacent to the measurement sites, and runs in a primarily west-east direction, offset about 18° towards a southwest-northeast direction. The highway is divided into collector and express lanes, with the inner eight lanes being east- and west-bound express lanes with few entrances or exits, and the remaining outer lanes being collectors with merge and exit lanes a few hundred metres up- and down-stream the measurement location. This site included two instrument locations: the first was a permanent roadside station on the south side of the highway that was frequently downwind the road; the second location was
a background sensor placed north of the highway, which was frequently upwind the road. The north site was designated as the background site based on predominant wind directions and the fact that this site featured a temporarily deployed low-cost sensor platform, while the south site features a permanent air quality station operated by the Ontario Ministry of the Environment, Conservation and Parks. Figure 1 maps the city sites in detail.





**Figure 1: Top: locations of measurement sites throughout Toronto region. Bottom: detailed map of the Highway 401 field study site. Inset bottom: wind rose measured at Highway 401 roadside (downwind) station during the study period. Throughout this document, the Highway 401 downwind roadside station is referred to as "highway roadside downwind" or "highway downwind", and the Highway 401 upwind background site is referred to as "highway upwind background" or "highway upwind".**

In addition to the primary highway site, we recorded pollution concentrations at three additional sites throughout the Toronto area. The first site was the Wallberg urban near-road site, located at the University of Toronto's Wallberg Memorial Building at UTM 629381 m E 4835252 m N 17N. This site features a similar set of air pollution instruments as the permanent Highway



401 downwind site, and was located 15 m from a major urban road and about 40 m from an intersection. The remaining two sites were designated as distant urban background sites, not near any emissions sources of comparable magnitude to Highway 401. The first urban background site was Downsview, located at UTM 623330 m E 4848631 m N 17N. This site is in a green space near an office building and is about 175 m from the nearest road. The final site was the Hanlan's Point urban background station, located at UTM 630025 m E 4830061 m N 17N. This site is located on an island in Lake Ontario, south of Toronto's downtown core. The Hanlan's Point site is isolated from any nearby sources, with the only notable emissions source being a regional airport over a kilometre to the north.

All sites listed here except the highway upwind background site were equipped with a similar set of air contaminant instruments, detailed in the next section.

### 2.1.2 Airborne pollutants, traffic, and meteorology

We employed a variety of instruments to measure air pollutant concentrations, meteorology, and traffic counts. The instruments deployed at each site except the highway upwind background are listed in Table 1. We selected $NO_x$ ($NO + NO_2$), CO, $PM_{2.5}$, and $CO_2$ to cover a range of dominant sources: we expect $PM_{2.5}$ and $CO_2$ to have large regional background concentrations while CO and $NO_x$ are more sensitive to proximity to sources. For $PM_{2.5}$, given the dominance of regional transport and secondary formation, and the consequential homogeneity of this pollutant's concentration across urban areas, we expect that differentiating between local and background pollution might be difficult. However, we retained $PM_{2.5}$ to serve as a counterexample to the other pollutants, which have greater differences between local and background concentrations.

**Table 1. Air pollution, meteorology, and traffic count instruments deployed at each measurement site except the highway upwind background site.**

| Measurand | Symbol | Method | Instrument name | Manufacturer |
|---|---|---|---|---|
| Nitrogen oxides | $NO, NO_2, NO_x$ | Chemiluminescence | 42i | |
| Carbon monoxide | CO | Infrared absorbance | 48i | Thermo Fisher |
| Fine particulate matter | $PM_{2.5}$ | Nephelometry and beta attenuation | 5030(i) SHARP** | |
| | | Spectrometry | T640** | Teledyne API |
| Carbon dioxide | $CO_2$ | Non-dispersive infrared | LI-840A | LI-COR Biosciences |
| Onsite meteorology | $T, P, RH, u, \theta$ | Various | WXT520 | Vaisala |
| Traffic counts* | $N_{LDV}, N_{MHDV}$ | Radar | Smartsensor 125HD | Wavetronix |

*Traffic counts were only recorded at the Highway 401 downwind site, and only for the nearest eight lanes. *LDV* = light duty vehicles, *MHDV* = medium and heavy duty vehicles.
**$PM_{2.5}$ at the Hanlan's Point background station was measured with a Teledyne API T640 while other sites used the Thermo Fisher 5030 or 5030i SHARP.

We acquired additional micrometeorological measurements for dispersion models from various sources, which we detail in the appendices. We used dispersion model outputs as exogenous variables for regression methods, described in more detail in Appendix C. At the Highway 401 north background site, we deployed a low-cost AirSENCE air pollution measurement system (AUG Signals, Toronto, Canada). This system utilizes a variety of low-cost sensor systems to simultaneously measure a variety





of pollutants, including $NO_x$, CO, $CO_2$, and $PM_{2.5}$. Morris et al.(2020) has previously explored the performance of the AirSENCE system.

For $PM_{2.5}$ at the Hanlan's Point site, we collected concentrations measured with the Teledyne API T640 rather than the Thermo Fisher SHARP instrument deployed at each other site (also again excepting the low-cost instrument upwind the highway). Zheng et al. directly compared two T640s to the same model SHARP used here, and reported variations up to 3 to

5 $\mu g \cdot m^{-3}$ in concentration ranges similar to those typically measured here, with the T640 more often reporting slightly higher concentrations than the SHARP. The possibility that $PM_{2.5}$ measured at Hanlan's Point may be slightly inflated should be kept in mind when reading results that directly compare concentrations across sites. Presumably, the low-cost sensor-based $PM_{2.5}$ we measured north of the highway also deviated from reference instruments by similar or larger amounts, however as explained below, we produced a corrective calibration for the low-cost sensor platform prior to deployment. We also found that when

directly comparing hourly $PM_{2.5}$ concentrations between SHARP and T640 instruments across sites used in this study, variation between instruments was similar to variation between sites, suggesting no systematic bias due to instrument differences. Should any disagreement between instruments exist anyways, this should only affect our results in cases where measured concentrations are compared directly – in cases where data were included in regression models, any offset in measured concentration should have a limited impact on regression results, as such offsets can be accounted for via intercept and

regression coefficients in linear models, or through similar underlying mechanisms in the machine learning model we applied here.

We averaged concentration and meteorology measurements to the nearest minute. To ensure the low-cost AirSENCE instrument platforms reported concentrations comparable with reference instruments to the greatest extent feasible, we applied multiple quality control and calibration steps prior to analysis. In particular, we addressed calibration and drift in some of the

low-cost sensors through comparison with other sites, and corrected the low-cost $PM_{2.5}$ measurements for hygroscopicity with the correction procedure devised by Crilley et al. (2018). We also placed the AirSENCE device atop the downwind highway station for nearly 18 days at the start of our measurement campaign, and used this co-location period to calibrate the AirSENCE's sensors against the station's reference instruments, controlling for interference from humidity, pressure, and temperature. We describe these preprocessing steps in greater detail in Appendix B.

Additional information on some of these same sampling sites and instruments can be found in publications by Wang et al. (2018), Hilker et al. (2019), and Jeong et al. (2020); this list is not exhaustive and these sites have been employed in a variety of prior air pollution studies.

## 2.2    Separating measured local and background concentrations at the highway site

To choose when we could consider the difference between near-road and upwind measurements as local concentrations,

$C_{local}$, we considered the relationship between measured concentrations and wind at the highway site. From Figure F.1 we identified which wind directions to subsample from our measurements to isolate local and background signals: we selected periods where wind direction relative to the road was between 80 degrees to the northwest and 40 degrees to the northeast.





The asymmetry in downwind directions relative to the road could be explained by traffic-induced turbulence, which can influence bulk air flow above the road (Hashad et al., 2022); since station south of the highway is nearest an eastbound lane, those lanes might add a westerly component to the observed wind direction. From Figure F.2 we also observed that some downwind roadside ($C_{meas}$) and traffic-related ($C_{local} = C_{meas} - C_{bkg}$) concentrations diverged below wind speeds of about $1.0 \text{ m} \cdot \text{s}^{-1}$. At low wind speeds, measurement of wind direction becomes unreliable, so identifying up- and down-wind periods is not possible with stagnant winds. Further, at low wind speeds the likelihood of vehicle-induced turbulence effecting the background measurements increases. To avoid analysing the lowest wind speed periods where these issues might be prevalent, we subsampled $C_{bkg}$ measurements for periods where concurrent wind speeds were $\geq 1.0 \text{ m} \cdot \text{s}^{-1}$ in addition to the requirement for concurrent wind directions falling within the above-mentioned range.

When applying measurements or estimates of background concentrations, in some applications it would be useful to further limit valid measurements of $C_{bkg}$ to periods where $C_{meas} \geq C_{bkg}$. If the reverse were true, this would imply $C_{local}$ is negative; physically, a negative $C_{local}$ might indicate that emissions from the source of interest are low and that upwind pollutant concentrations have diluted between the upwind and downwind sensors, and any emissions from the source of interest are not large enough to overcome this dilution. A negative $C_{local}$ could also be caused by a source near the upwind $C_{bkg}$ measurement that is not or only partially captured at the downwind site. In practice, negative measures of $C_{local}$ might not be useful. For example, in applications where $C_{local}$ is used to calculate emission rates from the source of interest, a negative value would imply absorption/reduction rather than emission, which in many cases would be impossible. In our analysis here we chose not to remove periods where $C_{meas} < C_{bkg}$ to avoid eliminating too great a portion of our measurements from our analysis, and to acknowledge that for pollutants where background concentration makes up a large portion of the whole measured concentration (as is the case for $CO_2$ and $PM_{2.5}$), the difference between $C_{meas}$ and $C_{bkg}$ can be small enough that instrument sensitivity will play a role in determining if the difference between the two is measurable.

We provide some additional discussion on the relationship between wind speed and $C_{local}$ in Appendix F.

## 2.3 Predicting background concentrations at the highway site

### 2.3.1 Onsite background concentration ($C_{bkg}$) prediction methods

We tested here nine methods of estimating background concentration measured upwind the highway: two distant urban background stations, three frequency methods, three regression methods, and a final ensemble method.

The distant urban background stations we tested for estimating $C_{bkg}$ were the same two urban background stations mentioned previously:

- Downsview station, located in an urban area but 175 m from the nearest road (Site C in Figure 1).
- Hanlan's Point station, located on an island in Lake Ontario, isolated from any nearby emissions (Site D in Figure 1).



We tested three frequency methods, including one mentioned in the introduction:

- A naïve rolling minimum, with the length of the rolling window optimized to minimize prediction error. We did not expect this method to produce accurate predictions, but included it as minimally simple approach.
- The pseudo-wavelet method devised by Wang et al. (2018).
- A rolling ball background subtraction – rolling ball algorithms are common in image processing, where they are used to correct unevenly intense image backgrounds. To our knowledge, this is the first case of rolling ball algorithms applied in air pollution research.

We also included three regression methods:

- Traditional ordinary least squares (OLS) multi-variable linear regression.
- Regularized (elastic net) regression, which is a linear model with two regularization terms that can be optimized to control for regression prediction overfitting.
- Machine learning regression with XGBoost – this model can produce accurate non-linear predictions and has a large number of hyperparameters that can be tuned to control overfitting, degree of variable interaction, model complexity, etc. The XGBoost model has been successfully deployed previously in air quality studies, demonstrating its potential usefulness (Xu et al., 2020b, a). We manually set some hyperparameters and optimized others to minimize error. See Appendix C for details on how we specified XGBoost regressions.

For each regression method, we included a variety of predictive covariates in addition to concentration measured downwind of the road. These covariates included concentrations measured at the distant urban background stations, vehicle counts split by vehicle size, predictions of pollutant dilution from the RLINE dispersion model, and meteorology measured at the Highway 401 site (Snyder et al., 2013). In some cases, we transformed covariates prior to fitting regression models to increase the linearity of the relationship between covariate and measured $C_{bkg}$, and for regression models we scaled predictors to mean zero and unit variance prior to fitting. We provide a detailed list of covariates in Appendix C.4.

Finally, we included one additional ensemble model: this final method was a regularized (ridge) regression using the predictions from each of the prior listed methods as inputs, and optimized to minimize root mean squared error in cross-validated predictions. This ensemble model was the most complex approach we employed and was fit last because it required the outputs of each prior model.

Extended descriptions of each of the algorithmic methods for estimating background concentration (i.e. all methods listed here except the urban background stations) are provided in Appendix C.

### 2.3.2 Optimizing prediction methods and evaluating accuracy

Many of the above methods for predicting $C_{bkg}$ require user-specific parameters. To select these parameters, we applied a similar process across each method. For each algorithmic method, we varied input parameters iteratively or semi-randomly via Bayesian hyperoptimization and selected the parameter that produced the lowest prediction error (Akiba et al., 2019). In





each case we evaluated prediction error with five-fold cross-validation to control for overfitting. The only exception was OLS, which has no hyperparameters to tune, however we still evaluated its accuracy with the same cross-validation scheme.

To evaluate model accuracy, we calculated a variety of metrics using a similar approach as when optimizing. These metrics included root mean square error (RMSE), mean absolute error (MAE), coefficient of determination ($R^2$), fractional bias (FB), among others. Additional details on $C_{bkg}$ prediction method optimization and evaluation, including details on optimized hyperparameters, cross-validation, and metrics are included in Appendix C.

## 3    Results and discussion

### 3.1    Geographic variability of urban background concentrations

After defining when a measurement is considered background at the highway site, we first compared average background concentrations at the three sites in the Greater Toronto Area. Figure 2 summarizes average concentrations while Figure 3 depicts their diurnal patterns. From these figures, we can directly compare typical levels and daily patterns in background concentrations across a city. Table 2 quantifies geographic and temporal variability in local and background concentrations at the same sites.

**Table 2: Mean and standard deviations (s.d.), and coefficient of variation (c.v. = s.d./mean) of pollutants measured at each study site, and means and standard deviations of differences between selected sites. The HWY Down – HWY Up row is the difference between up- and down-wind at the Highway site, summarizing variability in local ($C_{meas} - C_{bkg}$) concentrations. The Downsview – Hanlan's row is the difference between Downsview and Hanlan's Point sites, capturing geographic variability in backgrounds. Values rounded to two significant figures.**

|  | CO [ppbv] | | | $CO_2$ [ppmv] | | | $NO_x$ [ppbv] | | | $PM_{2.5}$ [µg·m$^{-3}$] | | |
|---|---|---|---|---|---|---|---|---|---|---|---|---|
|  | Mean | s.d. | c.v. | Mean | s.d. | c.v. | Mean | s.d. | c.v. | Mean | s.d. | c.v. |
| Highway downwind roadside | 380 | 160 | 0.42 | 460 | 30 | 0.064 | 45 | 33 | 0.74 | 6.4 | 5.6 | 0.87 |
| Highway upwind background | 230 | 120 | 0.54 | 440 | 30 | 0.068 | 16 | 18 | 1.1 | 4.8 | 4.7 | 0.99 |
| Downsview | 220 | 97 | 0.43 | 450 | 23 | 0.053 | 15 | 17 | 1.2 | 6.5 | 6.2 | 0.95 |
| Hanlan's Point | 220 | 62 | 0.28 | 440 | 15 | 0.034 | 7.9 | 11 | 1.3 | 6.3 | 4.7 | 0.75 |
| Wallberg (Downtown) | 240 | 85 | 0.36 | 450 | 20 | 0.044 | 14 | 12 | 0.9 | 5.9 | 4.8 | 0.82 |
| HWY Down – HWY Up | 150 | 110 | 0.69 | 17 | 17 | 0.99 | 28 | 25 | 0.9 | 1.8 | 3.9 | 2.2 |
| Downsview – Hanlan's | 9.9 | 84 | 8.5 | 8 | 19 | 2.4 | 6.9 | 13 | 1.9 | 0.19 | 4.5 | 24 |

*The lower $PM_{2.5}$ at highway upwind background only included periods where the sensor was upwind (northerly) of the road, whereas other sites were not restricted by wind direction or speed. When limiting considered winds across sites, $PM_{2.5}$ backgrounds at other sites were more comparable to the highway upwind background site (Figure 5).

For CO, $CO_2$, and $NO_x$, we recorded the greatest average concentrations at the Highway 401 downwind site; for $PM_{2.5}$ where it was second-greatest at the Highway 401 downwind site. High concentrations downwind the road is sensible given the intensity of traffic on this road. For example, the ratio of downwind/upwind concentration was greatest for $NO_x$: median total downwind $NO_x$ was 2.75 times greater than upwind background $NO_x$ at the highway site, which is unsurprising since $NO_x$ is





generally understood to be a strong traffic tracer. In the context of Figure 2, background $NO_x$ appears similar between the highway, Downsview, and Hanlan's sites, however this is misleading: low average background $NO_x$ concentrations mean that the percent differences between sites are relatively greater than for pollutants like $PM_{2.5}$ and $CO_2$, which have large backgrounds. In other words, the small concentrations of background $NO_x$ make for small absolute differences in background 330 concentration but larger ratios between sites. This introduces a contradiction in measuring background $NO_x$: background concentrations are low compared to near-source concentrations, so assuming a low or zero background would induce less error than such an assumption would induce for pollutants like $CO_2$ or $PM_{2.5}$, but at the same time assuming homogenous background $NO_x$ concentration would create the greatest percent error between background sites. The fact that total $NO_x$ concentrations are nearer zero than other pollutants increases the overall sensitivity of the difference between $C_{meas}$ and $C_{bkg}$, making it 335 paradoxically more sensitive but easier to estimate. Effectively the reliability of background estimation required differs for different applications. Even a rough estimate of the $NO_x$ background will be adequate when the application is subtracting this small value from a much larger total $NO_x$ concentration measured roadside (downwind). In contrast, it is challenging to evaluate how background $NO_x$ differs between locations, given that the background $NO_x$ concentrations will be small and difficult to estimate reliably. This is reflected in Figure 3, where diurnal background $NO_x$ measured at the Hanlan's Point site 340 is never equal to the other two background sites, whereas $CO_2$ and CO had similar concentrations across all sites during at least some times of the day. On the other hand, like CO and $CO_2$, there was some correlation in diurnal trends between background $NO_x$ recorded at the highway and Downsview sites.





**Figure 2. Box-and-whisker plots of concentrations measured at the various sites throughout Toronto. Hatched boxes indicate sites near and/or downwind a road (i.e. non-background sites). Boxes extend to 25th and 75th quantiles, whiskers extend an additional 1.5 interquartile ranges. Middle bars are medians.**





For CO, we measured similar average background levels at the Highway 401 downwind site and the Downsview urban background site, with the largest deviations between the two occurring during morning rush hour (Figure 3). There are two possible explanations for this morning divergence: first, higher nearby anthropogenic activity and thus emissions coupled with lower wind speeds in mornings would increase heterogeneity in urban background concentrations across the city. Second, during low morning wind speeds, emissions from the highway might reach the background station. However, we subsampled our highway upwind background measurements for periods with non-stagnant winds, so this second explanation should have a limited effect on our measurements; thus the morning rush-hour background CO differences in Figure 3 indicates increased spatial background heterogeneity during these times. CO measured at the Hanlan's Point urban background station were fairly level throughout the day, with a possible slight peak during morning rush hour; CO at Hanlan's Point was roughly 5 to 25% lower than the backgrounds measured elsewhere in the city, except during midday to early afternoon when concentrations were lowest and similar at all three sites. The correlation between CO measured at the highway upwind background and Downsview suggest that the Downsview site, situated 9.8 km northeast of the highway site, may serve as a good estimate of background CO concentrations with only a linear adjustment (e.g. the slope between Highway 401 and Downsview is < 1 in Figure J.1 but correlation is strong and intercept is near zero). At the Highway 401 site we measured background concentrations only when the sensor was upwind the road; further upwind was a suburban residential area north of the highway, so emissions from gas-fuelled furnaces may compound the background heterogeneity from low morning wind speeds we mentioned previously, especially given that our measurement campaign took place during winter months.

Like CO, background $CO_2$ concentrations had correlated diurnal trends and levels at the highway and Downsview locations, with higher rush-hour concentrations at the highway. This is indicative of spatial heterogeneity in $CO_2$ concentrations across the city, especially during mornings, as we observed for CO. The difference in average concentrations between the near-road and urban background sites is further notable given that we aligned baseline $CO_2$ concentrations at all sites during measurement preprocessing. Accordingly, the remaining differences indicate the near-road sites measured more transient high $CO_2$ concentrations, which in turn suggests non-constant sources upwind of these sites. The differences between $CO_2$ measured at the urban background stations and the highway upwind background means those distant urban background stations would not serve as adequate estimates of background $CO_2$ at the highway site if considering minutely or hourly data; conversely, the similarity in overall average background $CO_2$ concentrations suggests that if we were to consider only long-term (i.e. 24 h or greater) averages, distant urban background stations provide reasonable estimates of average background $CO_2$ concentrations, especially when comparing the backgrounds at the highway and Downsview sites (Figure 2 and Table 2).

The only notable feature in diurnal patterns of $PM_{2.5}$ background concentrations was a shallow noon-to-early-afternoon valley at Downsview and Hanlan's Point, which can be explained due to evaporation of morning-emitted ammonium nitrate, a precursor to $PM_{2.5}$, due to higher midday temperatures. The Highway 401 background sensor recorded the lowest median concentration, but differences in $PM_{2.5}$ concentrations across the city generally did not appear significant; in other words, we found $PM_{2.5}$ was spatially homogeneous across Toronto (Figure 2). This may be reflective of dominant sources and processes contributing to particulate matter in Toronto. Lee et al. (2003) observed over two decades ago that secondary processes were




a major source of total $PM_{2.5}$ in Toronto, while more recently Jeong et al. (2020) showed that, while source profiles have changed in the intervening years, secondary sources remain dominant. The importance of such secondary formation processes coupled with the trends in Figure 2 and Figure 3 indicate that separating the contributions of background concentrations and

primary emissions to $PM_{2.5}$ concentrations might not be feasible using time-series (frequency) and regression methods such as those discussed here. Conversely, homogeneity of $PM_{2.5}$ concentrations means urban background stations will provide a good estimate of background $PM_{2.5}$ throughout the city if we consider only long-term averages, like our observation of homogenous long-term averages for $CO_2$.


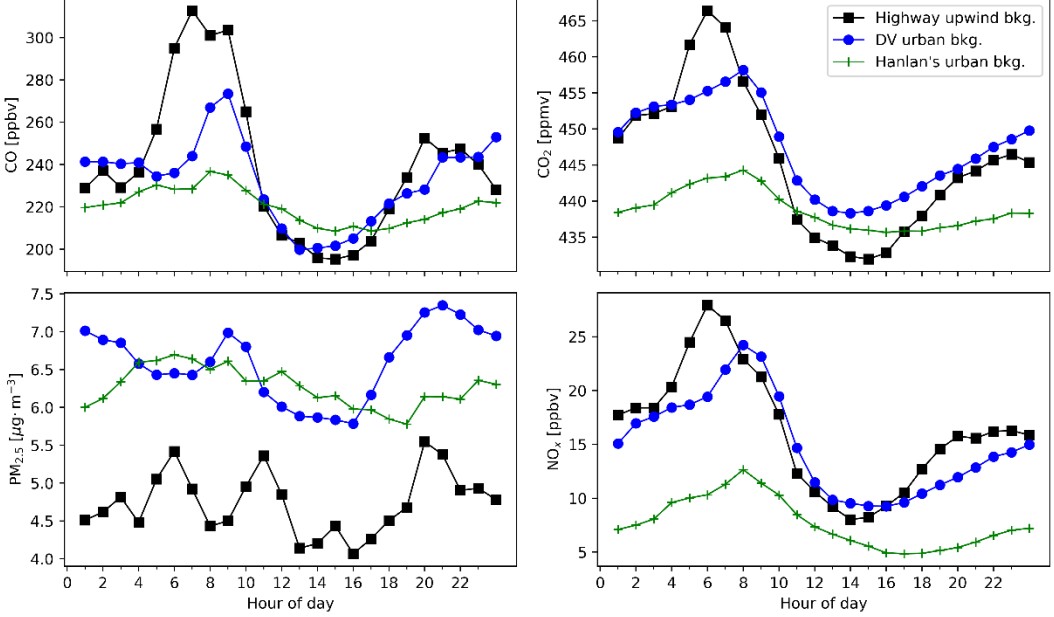

**Figure 3. Hourly mean diurnal profiles of measured background pollution concentrations at three stationary measurement sites in Toronto. For the Highway 401 site, these figures depict measurements from the background sensor only during periods where the background sensor was upwind the road and wind was not stagnant, producing a valid measure of $C_{bkg}$ as defined in the**
**methodology. Downsview (DV) and Hanlan's backgrounds had no wind direction restrictions.**

For CO, $CO_2$, and $NO_x$, the correlation in diurnal patterns between background concentrations measured at the highway and Downsview sites suggests that the Downsview station, situated within the city but about 175 m distant from the nearest notable traffic emissions source, may serve as an adequate estimate of upwind concentrations for measurements near sources like the highway in Toronto, but that the accuracy of this estimate would be reduced during mornings and evenings, when spatial

heterogeneity across the city in background concentrations may be larger. A similar conclusion can be drawn from the values in Table 2, where highway upwind background and Downsview have similar average concentrations, but the highway upwind background tended to have greater standard deviations and/or greater coefficients of variation, except for $NO_x$ which had



similar coefficients between highway upwind background and Downsview. Conversely, concentrations measured at the Hanlan's Point background station, which reflects regional background concentrations, are likely to underestimate highway

upwind backgrounds both in terms of average levels and variation.

Across pollutants, the level of hour-to-hour variability in Figure 3 and standard deviations in Table 2 correlated with the proximity of sites to pollution emissions sources. The highway upwind background, while isolated from the road of interest via wind direction, was still located in a dense urban area with a variety of emissions sources and had strong diurnal patterns throughout the day. We observed less hour-to-hour variability at the Downsview and Hanlan's Point urban background

stations. The Downsview site measurements were closer in magnitude to the highway upwind background, but variability was lesser, especially during morning and evening. The Downsview station is separated from immediate sources but is still within a few hundred metres of emissions sources, while concentrations measured at the more isolated Hanlan's Point were lower than all other sites (except for $PM_{2.5}$). Hanlan's Point lays on an island in Lake Ontario south of Toronto – while there is an airport on the same island, its runway is over 1 km away. We posit the lower CO, $CO_2$, and $NO_x$ at Hanlan's Point can be

explained from an absence of nearby sources, while the similar $PM_{2.5}$ is explained by both the dominance of secondary and regional particle sources.

Figure 4 shows scatters of measured background $CO_2$ at the three background sites. Figure J.1 to Figure J.3 show similar plots for the remaining measured pollutants. These figures only show periods where backgrounds were concurrently measured at each site, and only show a random 20% subset of measurements to speed calculation of the KDE and lessen figure density.

From these scatters we can derive similar conclusions about the relationship between background concentrations at various sites across the city. As we observed in Figure 2 and Figure 3, background concentrations at the near-road site might be reasonably estimated for some but not all pollutants. We observed that CO and $CO_2$ measured at the Downsview urban background station were somewhat correlated with background levels measured at the highway – thus we expect concentrations measured at Downsview to be important covariates in regression models predicting highway $C_{bkg}$ for CO and

$CO_2$ – but we noted that the correlation between Downsview and Highway 401 background concentrations was less clear for $NO_x$. $PM_{2.5}$ concentrations were mostly homogeneous across the city, and thus appeared more strongly correlated in scatters (Figure J.3). Background $NO_x$ concentrations were the least comparable between sites (Figure J.2), corroborating our earlier observation that, despite having low concentrations, $NO_x$ background concentrations are paradoxically very spatially heterogeneous and have a high degree of source-specific contribution at near-source sites. These inter-pollutant differences

again point to their individual levels of spatial homogeneity/heterogeneity. From these results we can rank pollutants in order of increasing background concentration geospatial heterogeneity: $PM_{2.5} < CO_2 \approx CO < NO_x$. While $PM_{2.5}$ is clearly the most homogeneous and $NO_x$ the most heterogeneous, the distinction in variability between $CO_2$ and CO is less clear.





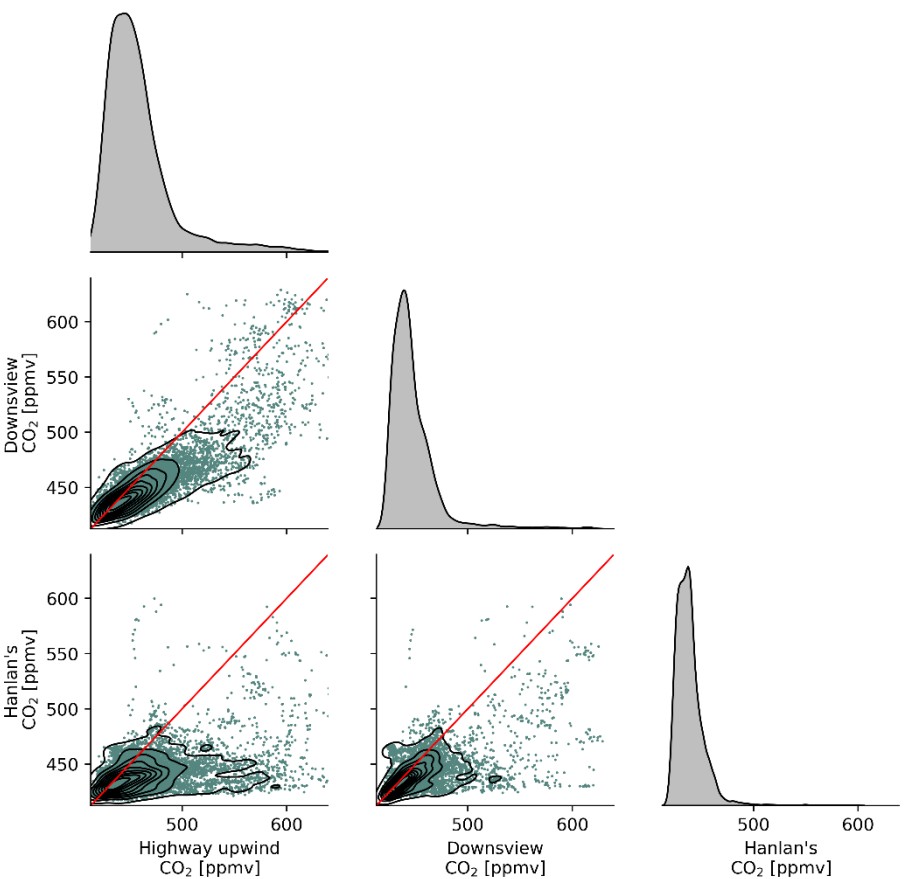

**Figure 4. Paired scatters and kernel density estimates (KDE) of measured background carbon dioxide concentrations at three stationary measurement sites in the Greater Toronto Area. HWY401 indicates the highway upwind background site, while other labels indicate their respective urban background sites. Red lines are 1-to-1.**

We also observed that this ranking of geographic variability was similar to the relative temporal variabilities in background concentration for each pollutant. The coefficients of variation for the difference between the Downsview and Hanlan's Point sites in Table 2 reflect a similar ordering, with the inter-site difference in PM$_{2.5}$ having the most variability relative to its mean, and NO$_x$ having the least – in other words, the coefficients of variation for inter-site differences express how much that pollutant varies temporally versus spatially, because a pollutant that varies over time but is spatially homogenous will have a large inter-site-difference coefficient of variability due to short-term concentration fluctuations or instrument noise, even if the coefficient for any individual site is low. We can also examine temporal variabilities and relative magnitudes of background concentrations indirectly by examining the best-optimized hyperparameters for the frequency methods, which we discuss in Appendix K.

From these comparisons of measured local and background concentrations, we can conclude that in some cases the urban background sites can provide a suitable estimate of highway upwind background concentrations, but for some pollutants and



times of day, a direct measurement or algorithmic estimate of background concentration is necessary. Accordingly, we further

applied and tested each of the background concentration prediction algorithms we introduced in the methodology.


## 3.2    Comparing performance of background concentration estimates

We subsampled roadside (downwind) and background (upwind) concentrations according to the definitions provided in the

methodology. We then optimized and estimated background concentrations using each of the frequency and regression

methods listed. Figure 5 shows diurnal patterns of measured and predicted concentrations at the Highway 401 site. The lines

for XGBoost and pseudo-wavelet show background concentrations estimated from the highway downwind data.

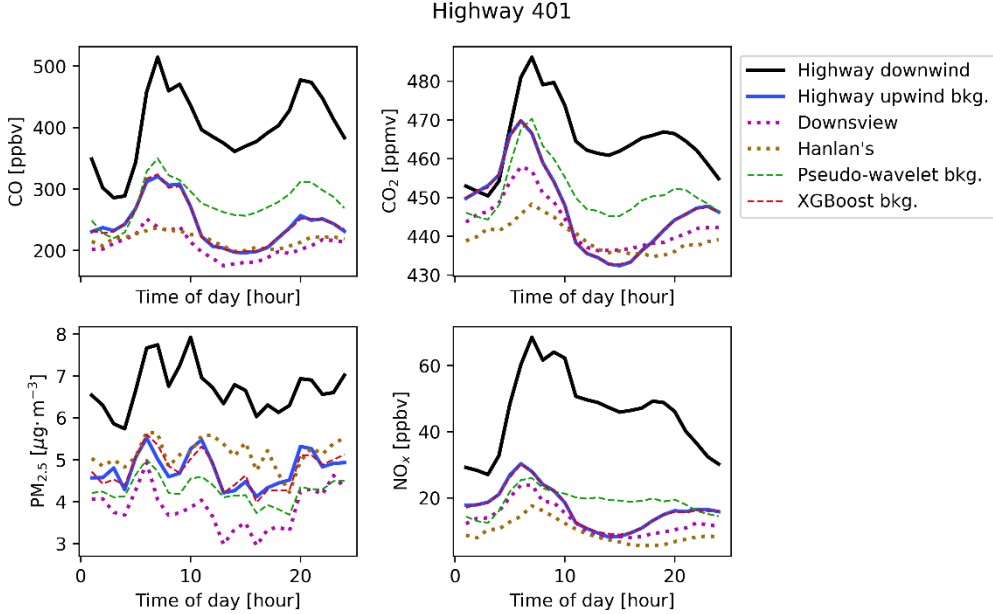

**Figure 5. Diurnal trends of measured total (black), measured background (blue) and predicted background (dashed purple and green, and solid red) concentrations at the Highway 401 site. Only periods containing valid measures of $C_{bkg}$ upwind of the highway as defined in the methodology were included in the highway upwind background data. Note that measured background**
**trends may differ slightly from Figure 3 as this figure only includes periods where all shown measured and predicted backgrounds were concurrently available.  The lines for "Highway upwind bkg." and "XGBoost bkg." are often superimposed yielding an apparently purple line.**

Figure 6 shows measured-predicted scatters for a selection of background concentration prediction algorithms. From these

scatters we observed that the accuracy of a method in estimating measured background concentrations was correlated with

model complexity – the computationally complex XGBoost model produced the most qualitatively accurate scatters of those

shown in Figure 6, while the simpler frequency (pseudo-wavelet) and urban background station (Downsview) estimates were

accurate at times but clearly less reliable than the XGBoost predictions.

For $PM_{2.5}$, we noted that our ability to produce an algorithmic estimate of measured background concentration was limited.

Poor accuracy of predictions is likely explained by the aforementioned sources and processes unique to $PM_{2.5}$ out of all the




pollutants studied here. For the remaining pollutants, accuracy varied between methods but appeared generally superior to that
of PM$_{2.5}$. However, as previously mentioned, this does not preclude us from viewing PM$_{2.5}$ as a counterexample by which we
can judge other, more accurately-predicted pollutants.

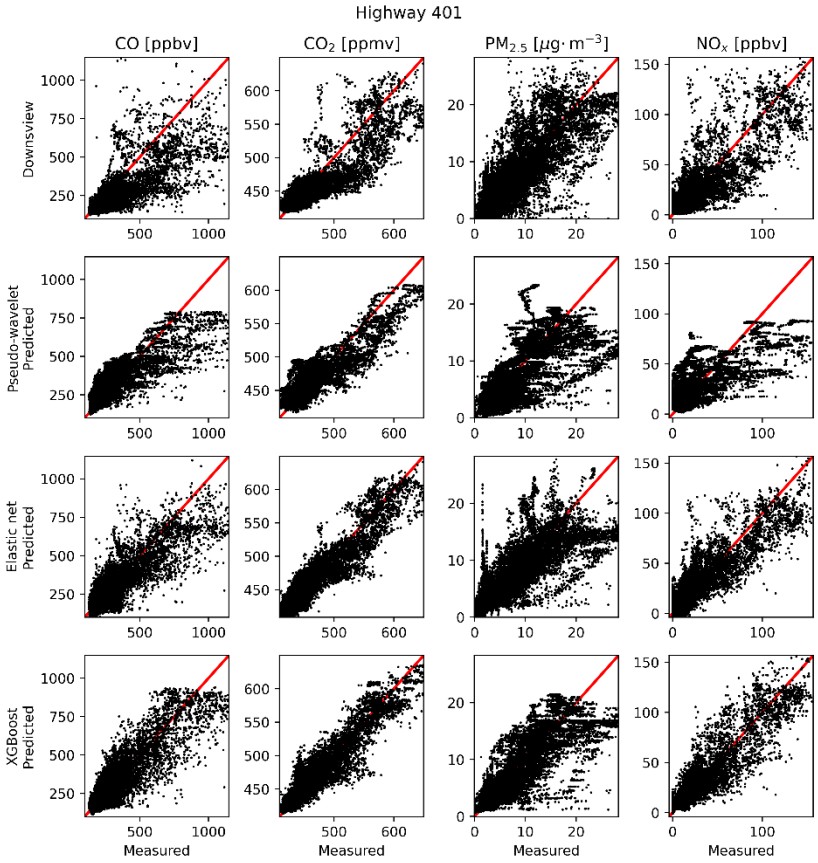

**Figure 6. Measured-predicted scatters for selected methods of estimating background concentration at the Highway 401 site.**
**Measured concentrations are true $C_{bkg}$ recorded by the AirSENCE device north and upwind of the highway. Scatters only include**
**periods where $C_{bkg}$ measures were valid as defined in the methodology.**

Figure 7 shows the root mean square error (RMSE) and coefficient of determination ($R^2$) of CO $C_{bkg}$ predictions using each

method, including the urban background stations, roughly ordered by increasing complexity and accuracy. Figure H.1 to Figure

H.3 show the same metrics for NO$_x$, CO$_2$, and PM$_{2.5}$. Where Figure 6 permits us to qualitatively examine $C_{bkg}$ prediction

accuracy, Figure 7 (and Figure H.1 to Figure H.3) quantitatively corroborate our observations that accuracy tended to increase

with model complexity. Unsurprisingly, the XGBoost and ensemble models generally had the greatest accuracy out of all

algorithmic methods, according to prediction RMSE and $R^2$. When compared with urban background stations, frequency

methods tended to have similar error to measured background data from Downsview in predicting $C_{bkg}$ (except for NO$_x$), and

regression methods, particularly XGBoost, had less error and greater $R^2$. OLS and elastic net had lower accuracy than XGBoost



models, indicating some degree of variable interaction or nonlinearity existed in background concentration behaviour, but the increase in accuracy from linear regression to machine learning was minor for all pollutants. Hanlan's Point always had greater error and lower $R^2$ than Downsview, a trend reflecting our above discussion on the suitability of using a distant urban background station for predicting onsite $C_{bkg}$. For $CO_2$ and $NO_x$ the incremental gain in prediction accuracy between frequency and regression methods was more apparent than for $PM_{2.5}$ and CO, suggesting accurate prediction of $CO_2$ and $NO_x$ might more strongly rely on information contained in predictors other than downwind $C_{meas}$. Interestingly, for $NO_x$ the predictive accuracy of frequency methods was worse than simply using measurements from the Downsview background station to predict $C_{bkg}$, suggesting background $NO_x$ cannot be extracted from downwind total $NO_x$ alone with high accuracy, although as previously discussed high accuracy is not needed for applications like resolving local contributions since background $NO_x$ is generally much lower than local $NO_x$. For every other pollutant the accuracy of the Downsview background station in predicting $C_{bkg}$ was worse or at least near that of frequency methods. This observation might also be reflective of our previously mentioned sensitivity in estimating background $NO_x$ due to its relatively low average concentrations.

For $PM_{2.5}$, the accuracy of algorithmic $C_{bkg}$ predictions did exceed that of the Downsview station, but the relative incremental gain in accuracy was less clear than for other pollutants, suggesting little benefit can be gained for algorithmically predicting background $PM_{2.5}$ over simply using an urban background station. Only the XGBoost and ensemble models had notably superior accuracy for $PM_{2.5}$, indicating that greater complexity is necessary to accuracy predict background $PM_{2.5}$ than for other pollutants. These trends broadly align with our prior discussion on the homogeneity and complexity of sources and processes governing background $PM_{2.5}$. However, the RMSE of the low-cost sensor placed upwind of the highway versus a reference sensor was greater than the mean difference between up- and down-wind $PM_{2.5}$ at the highway (Table B.1), suggesting that in addition to the homogeneity of $PM_{2.5}$ (Figure 2 to Figure 4), our ability to separate $C_{bkg}$ from $C_{meas}$ was limited for $PM_{2.5}$, which would inherently limit our ability to predict $PM_{2.5}$ $C_{bkg}$.



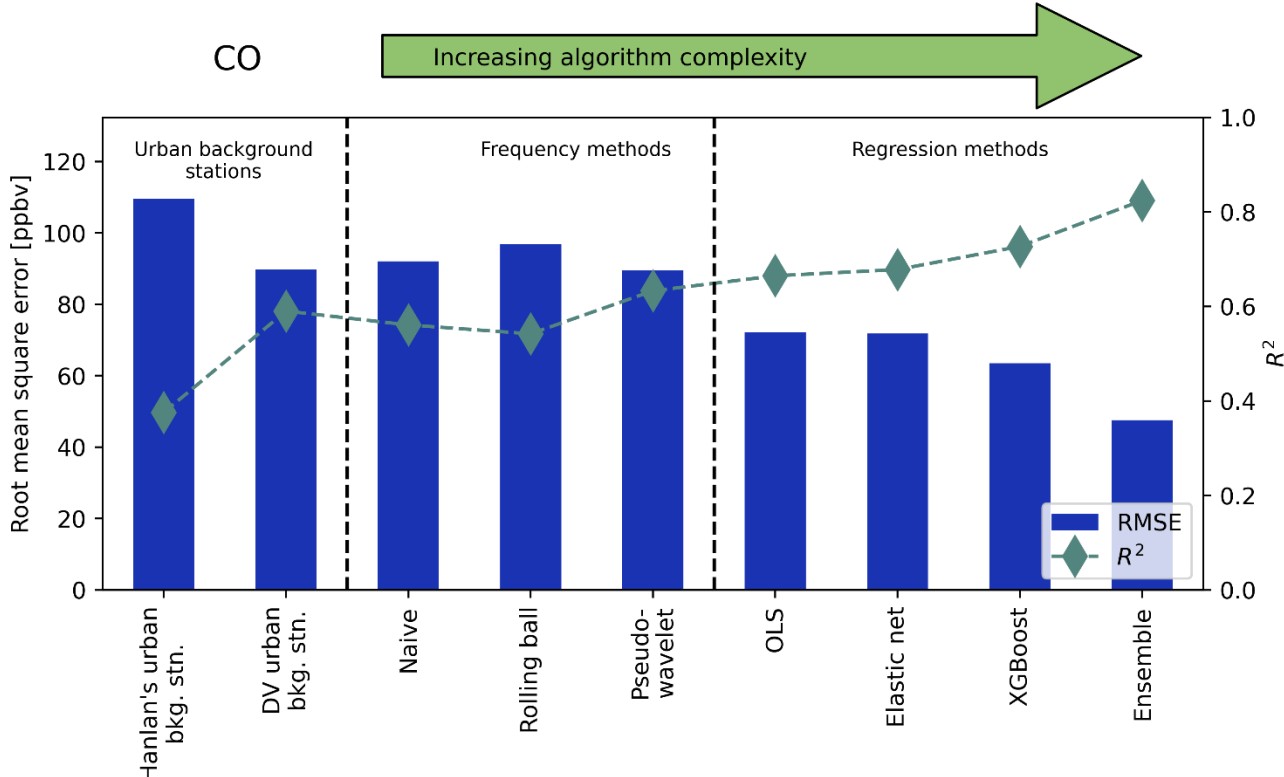

**Figure 7. Root mean square error (RMSE, bars) and coefficient of determination ($R^2$, diamonds) for predicted background CO at the highway site, as predicted by each method tested here. Scores show the accuracy of each method in estimating true upwind background concentration, with lower RMSE and greater $R^2$ being better. Scores were calculated as the mean across five-fold cross-validation.**

For CO and $CO_2$, there is some similarity in accuracy for frequency methods and regression methods. RMSE and $R^2$ for CO predictions from regression methods were only slightly better than RMSE for frequency methods. For $CO_2$, prediction RMSE and $R^2$ appeared to improve from frequency methods to regression methods, and again to the ensemble model, indicating similar levels of accuracy within each class of algorithmic prediction models.

## 3.3    Importance of site-specific covariates

We fit each method to only a single field study site, so it is difficult to conclude if our results are generalizable for urban background concentrations or if they are specific to the single highway site studied. However, we can gain some insight into the generality of our conclusions by testing the importance of site-specific information in producing accurate estimates of background concentrations with the regression methods tested here. Specifically, to test the importance of onsite information in predicting background concentrations, we refit our XGBoost model after shuffling covariates specific to the highway site, but XGBoost hyperparameters and the total number of variables remained unchanged. Shuffling covariates refers to the process



by which one input variable at a time is randomly shuffled so the measurements of that variable are no longer in order relative to other input and target features. By shuffling covariates and refitting, we remove possible correlations between site-specific features and the target measured background concentration but retain the same set of features so we can refit the XGBoost model without retuning hyperparameters, enabling comparison of XGBoost predictions with and without highway-specific inputs.

To produce this regression, we shuffled covariates specific to the highway emissions source, including RLINE dispersion estimates, highway traffic counts, and traffic-weighted average vehicle speed. The site-specific measurements we left unshuffled were downwind total concentrations, $C_{meas}$, the target upwind background concentrations, $C_{bkg}$, and meteorology. We chose not to shuffle meteorology based on our observation that meteorology is usually similar across the city at any moment in time and thus could feasibly be measured offsite; meteorological measurements are also often widely available or measurable with relatively low-cost instruments. Figure 8 shows normalized prediction errors for $C_{bkg}$ predicted via XGBoost for each pollutant with and without shuffling.

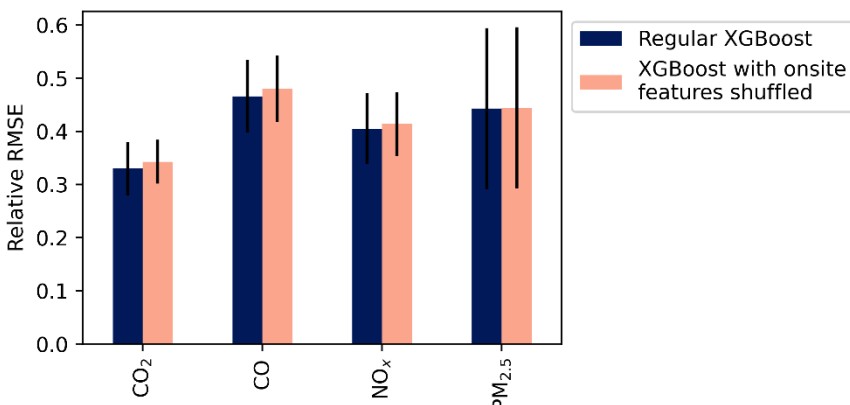

**Figure 8. Relative RMSE for XGBoost-predicted $C_{bkg}$ with and without covariates specific to the highway field site included in model. RMSE was calculated via five-fold cross-validation, relative RMSE is RMSE divided by the standard deviation of the target regressand ($C_{bkg}$). Whiskers are standard deviations across folds.**

The errors in Figure 8 suggest that removing information specific to the highway site did not produce a significant change in XGBoost model accuracy. The absolute percent difference between RMSE with and without site-specific variables shuffled was less than 5% for all pollutants, and differences were within one standard deviation across cross-validation folds, indicating little or no significant difference between models with and without shuffled site-specific variables. This indicates that most of the variability in $C_{bkg}$ was explained by highway downwind concentrations and other covariates not specific to the highway – it is also reflective of our observations in Figure 7 and Figure H.1 to Figure H.3 that predicting $C_{bkg}$ with concentrations measured at the Downsview urban background station, while less accurate than some other methods, still produced prediction $R^2$ exceeding 0.5 for all pollutants. Since concentrations measured at Downsview and the Highway were included as predictors in both cases in Figure 8, we can indirectly conclude that concentrations measured at Downsview coupled with concentrations



measured downwind the highway together contain most of the information necessary to accurately predict $C_{bkg}$, and that adding more emissions-source-specific covariates only marginally increased prediction accuracy.

This lack of difference might imply our model of background concentrations is not site-specific. That is, the XGBoost model without highway-specific covariates might be transferable to other locations. This in turn would mean that the spatial variation of background across the city is mostly encompassed within information provided by measuring the total concentrations at different sites, consistent with the assumption underlying frequency-based methods. With only one near-source site in this study, we did not further test this transferability. At the very least, this result shows our methodology might be successfully repeated at new near-source sites without requiring as many site-specific covariates as we included here.

### 3.4     Regression model feature importance

We can examine feature importance in the XGBoost models for each pollutant to gauge covariate importance for estimating $C_{bkg}$. We achieve this using Shapley Additive Explanations (SHAP) – SHAP plots can provide explanations of feature importance for complex nonlinear models where simple coefficients are not available, as is the case with XGBoost (Lundberg and Lee, 2017). Additional examples of SHAP values in the context of air pollution research can be found from Xu et al.

(2020a, b). Figure 9 shows SHAP beeswarm plots for the XGBoost model predicting highway upwind background $C_{bkg}$ for each pollutant.

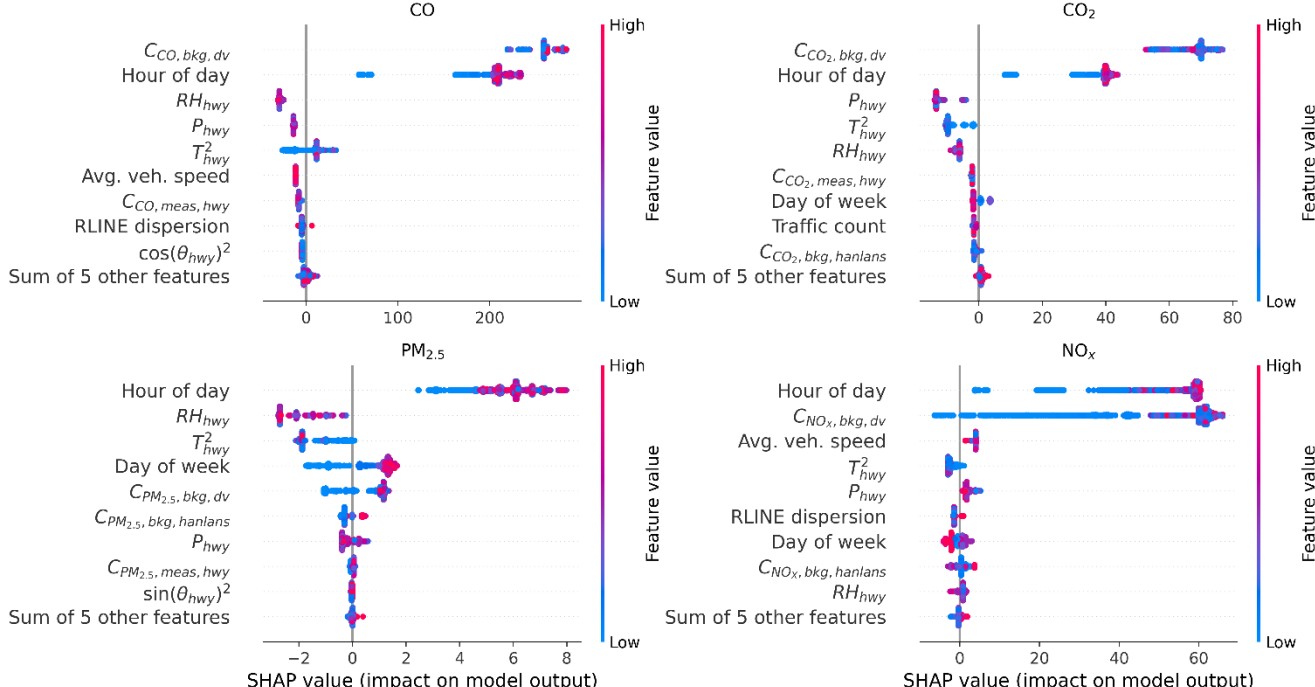

**Figure 9. SHAP beeswarm plots for XGBoost models predicting upwind background concentration at the highway site. XGBoost models were tuned via Bayesian hyperoptimization with five-fold stratified cross-validation. These figures indicate relative degree**



**of importance – for example, a blue dot far to the right on a feature indicates that a low value of that feature was associated with a high predicted concentration. Each dot represents one predicted concentration and one value of that feature. (bkg: background; dv:Downsview; hwy: highway; meas: measured)**

The SHAP values in Figure 9 suggest that for CO, $CO_2$, and $NO_x$, the most important predictors of upwind background at the Highway 401 site were concentrations measured at the Downsview urban background site and hour of day. This is

consistent with each of our prior discussed results: generally concentrations measured at Downsview can provide a fair estimate of mean background concentration levels, but these estimates may be inaccurate during some hours of the day and predictions can be notably improved through inclusion of additional information. The fact that time of day is an important predictor aligns with our observation that Downsview serves as a fair background estimate, except during morning and to a lesser degree evening rush hours (i.e. except during some hours of the day). Outside these important predictors, meteorology had notable

importances for all pollutants. Lastly, for $PM_{2.5}$ pollutant concentrations measured at Downsview, while still important, had a lesser impact on predictions, which is yet again reflective of the difficulty in predicting background $PM_{2.5}$ at the highway site.

Surprisingly, concentrations measured downwind the highway, $C_{meas}$, were much less important predictors in XGBoost than concentrations measured at Downsview. This was surprising both based on theory and when comparing against other methods: $C_{meas}$ should always be a direct sum of $C_{bkg}$ and local emissions, and thus we expect it to explain some variability

in background concentrations. This result was also in contrast to regression coefficients from our linear elastic net fits, which fit large coefficients on $C_{meas}$ for all pollutants (Figure L.1 to Figure L.4). Regardless, the results of this SHAP analysis suggest that $C_{meas}$ had a comparatively small impact on XGBoost predictions. On the other hand, we found that our frequency methods, which take only $C_{meas}$ as an input, had fair accuracy. These two results together suggest that to extract useful estimates of $C_{bkg}$ from $C_{meas}$, algorithmic methods benefit by considering not just concurrent measurements but past and

future values of $C_{meas}$ as well; in this study, only frequency methods considered lagged values of $C_{meas}$ when predicting $C_{bkg}$.

The importance of temperature for some predictions might be explained by an uneven distribution of measured temperatures. Most of our measurements occurred in winter with low temperatures, while a minority of measurements at the end of our study had higher temperatures. Because there are fewer samples with high temperatures, regression models risk placing a greater relative importance on those samples, inflating the relative importance of temperature. This can be improved upon in future

studies by extending a similar regression fitting approach to a longer measurement period.

### 3.5 Limitations of analysis

As this study examined only a single urban area, the applicability of our results to other urban areas relies on the assumption that many cities feature a similar variety and heterogeneity of emissions sources and geography. The sites explored here, including both urban background and near-road up- and downwind- sites, represented a variety of geographic features,

including proximity to a large body of water, green space, and proximity to emissions sources other than the road targeted at the highway site. While our analysis of XGBoost model accuracy without site-specific features in Figure 8 lends support to the idea that our model of background concentrations is not specific to the highway site, this conclusion is indirect and a better



method of testing transferability would be to apply our results at a new site with similar configuration of measurements up- and down-wind of an emissions source.

We also only explored background concentrations for four airborne pollutants: three gaseous and one particulate. For the gaseous pollutants tested, we expect that loss or formation via reaction will be low. While NO and $NO_2$ concentrations can vary rapidly near roads through reaction, we only considered the sum of the two, $NO_x$, which as a whole should not vary significantly through reaction over the distances from the highway investigated here. This simplicity of behaviour will simplify our models, and it is plausible that background pollutants with more complex reaction mechanisms or sources might require

more covariates to accurately predict with regression models. For example, modelling background ozone might benefit from including insolation as an exogenous predictor.

Lastly, it remains unclear if these models would transfer well to sites with different geometry, emissions sources, or typical weather. It is plausible that the strength of the methods tested here is due to the simplicity of the major source observed: the size and business of Highway 401 lends confidence to the assertion that it will be the dominant source of local airborne

pollution at the downwind highway site; traffic also has consistent diurnal patterns and emissions intensity is easily inferred through a simple traffic count. If the regression models presented here were refit near a source with different characteristics, such as an industrial source emitting at all hours of the day, or at a measurement site with multiple strong upwind sources, it stands to reason that predictive performance would be degraded.

## 4     Conclusions and recommendations

From the analyses and results presented here, we can draw conclusions on the behaviour and predictability of background concentrations and our ability to extract the influence of local emissions on measured air pollutions.

First, we observed that background air pollution concentrations varied geographically across the measured urban area during the study period. Variation was greater for $NO_x$ and CO, which are more strongly driven by localized emissions sources; variation was lesser for $CO_2$ and $PM_{2.5}$, for which background concentrations make up a greater portion of the measured whole,

and for which global or regional upwind emissions and processes dominate over localized emissions. This has implications for urban air quality studies targeting specific emissions sources, especially considering the growing popularity of deploying many low-cost sensors across a region to study the spatial spread of air pollutants. While placing a sensor near an emissions source might provide insight into the influence of that source on measured concentrations, the inhomogeneity of some background concentrations observed implies that researchers cannot always rely on distant measurements of background concentration to

isolate local emissions signals.

To address the geographic variability of background concentrations, we tested the extent to which urban or regional background stations can be used to predict background concentrations at a field site near a major urban highway. We found that concentrations measured at the distant background sites most accurately predicted highway upwind backgrounds for $CO_2$, and with slightly less accuracy for $PM_{2.5}$, CO, and $NO_x$, and was the most or second-most important predictor in regression



models for all but $PM_{2.5}$. This loosely aligned with our observation that $CO_2$ and $PM_{2.5}$ varied less across the city than CO and $NO_x$. It is intuitive that a distant background station would provide more accurate predictions of onsite $C_{bkg}$ for pollutants with low spatial variability in background concentrations, than for pollutants with higher variability. Our results also indicate that our low-cost instrument upwind of the highway produced useful measurements of background CO, $CO_2$ and $NO_x$ concentrations, provided it was calibrated via co-location with reference instruments prior to deployment. Comparison of

background concentration scatters and diurnal patterns indicate that the low-cost instrument was similarly capable of measuring these background concentrations as a reference station, despite its lower cost and weaker reported sensor sensitivity.

We tested seven algorithms for estimating background concentrations: three frequency methods using only measured roadside downwind concentrations as an input, three regression methods using roadside downwind concentrations and additional covariates as inputs, and a final ensemble model using the predictions from the prior six methods as covariates. We

observed that in some cases frequency methods outperformed distant urban background stations for estimating near-road $C_{bkg}$, in most cases regression models outperformed frequency methods, and the ensemble model always provided the best accuracy of all. This ranking of accuracy is broadly correlated with increasing complexity and number of inputs for each method: using an urban background station to estimate $C_{bkg}$ includes no information about the field study site or pollutant concentrations measured there, and this approach had lower accuracies and underpredicted backgrounds for some pollutants during rush hours.

Frequency methods included total measured roadside downwind concentrations, $C_{meas}$, and were somtimes superior to urban background stations. Regression methods incorporated additional covariates on top of $C_{meas}$ and outperformed frequency methods and distant background stations. Finally, the ensemble model, requiring the outputs of each prior method, had the greatest complexity and greatest accuracy.

Based on these results, we recommend that municipalities or air pollution specialists deploying sensors or monitors with the

aim of resolving the contribution of specific emissions sources consider carefully how they will measure or algorithmically isolate the contribution of background to total measured concentrations. Our measurement sites in Toronto reflected a variety of geographic features (varying built environments, proximity to water, extent of green space, etc.), indicating that our finding of varying background pollution concentrations might apply to other cities, given these features are common across many urban areas. From our analysis of background concentration prediction methods, we can provide recommendations for which

method users should choose based on their use-case and availability of data. These recommendations are loosely ordered by decreasing strength of predictive accuracy offset by decreasing requirements for instrumentation or additional measurement sites:

1.  If possible, direct measurement of background concentrations and winds immediately upwind the source of interest should always be preferred.

2.  In cases where measurements of $C_{bkg}$ upwind the source of interest are available for some but not all of the study period, we recommend applying a regression approach – these approaches had the greatest accuracy and our cross-validation results suggest they should extrapolate well to periods where $C_{bkg}$ was not directly measured. In





particular we suggest using XGBoost or similar machine learning approaches that allow for nonlinearity and interactions, which here had the greatest accuracy. However, we advise caution in applying regression models outside the conditions they were trained in, such as different sites or seasons.

3. For applications where only long-term averages (i.e. 24 h or longer) are of concern, using a distant urban background station as a proxy for true onsite $C_{bkg}$ measurements will prove sufficiently accurate, however for higher-resolution measurements, urban background stations may prove inaccurate during periods of peak emissions, such as during rush-hour near the highway emissions source studied here.

4. For applications where both upwind $C_{bkg}$ measurements and a suitable urban background station are both unavailable or too costly, we suggest applying one of the frequency methods described here, particularly the pseudo-wavelet method developed by Wang et al. (2018) or the rolling ball algorithm. For these frequency methods, in roadway applications we suggest using hyperparameters like those identified here in Table K.1. For pollutants other than those measured here, we suggest applying parameters like those in Table K.1 based on similarity in pollutant behaviour – for example, if a pollutant is expected to be a strong tracer or a local source, as NO$_x$ is for traffic, we suggest applying similar hyperparameters as used for NO$_x$ in this study.

5. In a similar vein, for cases where municipalities are deploying networks of sensors, or epidemiologists are exploring geographic variability of background concentrations vs. local emissions, we suggest applying the pseudo-wavelet or rolling ball frequency methods. While the context of our tests here were up- and down-wind differences targeting a single roadway emissions source, the theoretical basis of frequency methods – that background concentrations vary on a longer time-scale than local emissions – extends these methods to pollution concentrations regardless of proximity to one particular source. The pseudo-wavelet method applied in this context is also touched upon by Wang et al. (2018) and Hilker et al. (2019).

Generally, we do not suggest applying the naïve rolling minimum method tested here except as it was used here: as a bar by which we can judge other, better algorithms for predicting $C_{bkg}$, since the naïve rolling minimum can be considered the simplest but least accurate method for estimating background concentrations, so any other new method should outperform it. In a similar vein, usefulness of the ensemble model we tested is dubious – the extent of information and effort required to implement such a model for predicting $C_{bkg}$ seems to exceed the potential benefit of gains in predictive accuracy. Also, when choosing an urban background station to serve as a proxy for $C_{bkg}$, care must be taken to choose a station placed in an area with similar land use as upwind of the source of interest – a station like the isolated Hanlan's Point site we describe here would be unsuitable for sites surrounded by urban or suburban areas.

Finally, we suggest any study targeting specific emissions sources carefully consider how to extract local versus background contributions to measured concentrations, including but not limited to applying one of the methods tested here. We also encourage additional research in separating local and background concentrations, especially with different emissions sources, regions, or methods than those we explored here.





## 5    Appendices

### Appendix A        Micrometeorological and other inputs for RLINE

We used the RLINE model to produce dispersion estimates as an input feature for regression models in this study (Snyder
et al., 2013). The RLINE model uses outputs from the AERMET micrometeorological pre-processor produced by the United
States Environmental Protection Agency (U.S. EPA, 2004). AERMET requires a variety of micrometeorological
measurements as inputs, which can be provided in a variety of formats. We employed measurements from Toronto's Pearson
International Airport, acquired from the National Centers for Environmental Information Integrated Surface Database (n.d.);
and upper air measurements at Buffalo Niagara International Airport, acquired from the National Oceanic and Atmospheric
Administration's radiosonde database (n.d.).

We identified lane and receptor geometry using ArcGIS Pro and Google Earth Pro. We set initial vertical dispersion, $\sigma_{z,init}$,
using the recommended formula in the RLINE user manual, which in turn points to EPA guidance (Environmental Protection
Agency, 2010; Snyder and Heist, 2013). This formula uses vehicle heights and fleet mix to estimate initial dispersion; we
assumed vehicle heights of 1.5 m for light-duty vehicles and 4.15 m for medium- and heavy-duty vehicles, based on the same
EPA guidance document and the law in Ontario governing maximum vehicle height (Ontario, 2012). Other inputs were taken
from recommendations in the RLINE user manual.

### Appendix B        Data processing

To ensure air pollutant concentration measurements were accurate, realistic, and comparable between sites, we performed
an extensive quality assurance and control process on the raw measurements prior to use. First, gas-phase instruments at the
Downsview, Hanlan's, Wallberg, and Highway 401 south site are calibrated regularly.

Prior to analysis, we applied the following steps to raw measurements. For programmatic details, refer to the raw files
provided alongside this publication.

1.  We removed periods identified as invalid measurements in our measurement database for reasons such as
    calibration or maintenance. In some cases, we dropped additional measurements if it appeared the instrument was
turned back on too soon after calibration.

2.  We manually removed some periods that appeared to have extreme outliers or unusual behaviour suggestive of
    instrument malfunction, calibration problems, or transient spikes unrelated to the measured road emissions or
    background concentrations.

3.  We corrected $PM_{2.5}$ measurements from the AirSENCE instrument for interference from humidity with the
correction equation suggested by Crilley et al. (2018).

4.  We corrected for baseline drift in $CO_2$ measured at Hanlan's Point, Wallberg, and both Highway 401 stations by
    assuming concentrations measured at these sites must be similar to $CO_2$ measured at the Downsview site
    occasionally over a 48 h period. We selected the Downsview urban background station as the reference site for this
    adjustment because it was calibrated during the sampling campaign. We applied such a correction specifically by





730        calculating the rolling 48 h 0.5% quantile of each $CO_2$ signal and assuming these rolling quantiles must be equal –
           we then subtracted the difference between each site's rolling quantile and the Downsview quantile from the $CO_2$
           signal at each site (except Downsview, since it was treated as the reference).

   5.    We calibrated the Highway 401 background AirSENCE instruments by placing the sensor package on the roof of
           the Highway 401 south station for 17 days prior to deployment to the north side of the highway. With these 17
735        days' raw pollutant measurements, we calibrated the AirSENCE instrument against measurements from the south
           station's reference instruments. This calibration was specifically a linear regression, regressing a target function
           like:

$$C_{ref} = \beta_0 + \beta_1 C_{AS} + \beta_2 T + \beta_3 P + \beta_4 RH + \beta_5 C_{AS}T + \beta_6 C_{AS}P + \beta_7 C_{AS}RH \qquad (2)$$

           Where $C_{ref}$ are concentrations recorded by the reference instruments, $C_{AS}$ are concentrations measured by the
740        AirSENCE low-cost platform, $T$ is ambient temperature, $P$ is ambient pressure, $RH$ is ambient relative humidity,
           and $\beta$ are regression coefficients. We regressed this function for each pollutant, and then created predicted values
           of $C_{ref}$ for the entire measurement campaign, and treated these values as calibrated measurements from the
           AirSENCE device after we deployed it to the north (background) side of the highway.

   6.    After the above steps, we set concentrations less than zero to $10^{-5}$ for each pollutant. We applied this adjustment
745        to simplify analyses that required taking the logarithm of concentrations.

Table B.1 shows some measures comparing AirSENCE pollutant concentrations to reference instruments at the Highway
401 south station before and after calibration. These measures generally appear to indicate that the AirSENCE reported similar
concentration measurements to the reference instruments during the training period after measurements were preprocessed
using steps 1 through 5 above.





**Table B.1: Statistics comparing concentrations measured by the low-cost AirSENCE sensor platform to reference instruments before and after calibrating the AirSENCE measurements.**

| | Pollutant | $R^2$ | RMSE | $A_{F=1.1}$ | FB |
|---|---|---|---|---|---|
| Pre-calibration | CO | 0.92 | 82 | 0.18 | -0.19 |
| | $CO_2$ | 0.83 | 12 | 1.0 | -0.015 |
| | $PM_{2.5}$ | 0.75 | 4.2 | 0.16 | 0.19 |
| | $NO_x$ | 0.98 | 33 | 0.0025 | -0.56 |
| Post-calibration | CO | 0.93 | 36 | 0.77 | ~0 |
| | $CO_2$ | 0.85 | 8.7 | 1.0 | ~0 |
| | $PM_{2.5}$ | 0.78 | 3.6 | 0.20 | ~0 |
| | $NO_x$ | 0.98 | 4.5 | 0.58 | ~0 |

The performance statistics in Table B.1 imply that, after calibration, measurements captured by the low-cost AirSENCE sensors were comparable to those captured by the reference instruments, with small errors and effectively no bias for CO, $CO_2$, and $NO_x$. However, for $PM_{2.5}$, the fraction of values falling within a factor of 1.1 ($A_{F=1.1}$) and the RMSE imply that $PM_{2.5}$

measurements were relatively less accurate than other pollutants. This likely compounded with our observation of homogeneity in $PM_{2.5}$ background concentrations in the Toronto region, further reducing our ability to separate $C_{bkg}$ from $C_{meas}$ for $PM_{2.5}$ at the Highway 401 site. Accordingly, and as mentioned in the main article body, our ability to extract meaningful results at the Highway 401 site was lesser for $PM_{2.5}$ than for other pollutants. However, our observation that $PM_{2.5}$ was largely homogeneous across Toronto remains valid, as the low-cost AirSENCE device was only deployed at the highway upwind

background site.

**Appendix C       Descriptions of background concentration prediction algorithms**

The following sections list the various frequency- and regression-based algorithms we tested for estimating on-site upwind background concentrations. Most methods follow a similar optimization scheme, and all were tuned to produce the best

possible estimate of measured background, $C_{bkg}$, at the highway upwind background site.

Except where otherwise noted, we applied a similar optimization method for tuning and fitting each of these algorithmic models. We employed the *optuna* Python library, which applies Bayesian hyperoptimization to search the possible space of hyperparameters for an optimal configuration (Akiba et al., 2019). For scoring during optimization, we calculated the five-fold cross-validated root mean square error (RMSE) of predictions. In stratified cross-validation, the model is fit or regressed to

most of the data (the training set) while a subset is held aside (the test set); after fitting, predictions are generated for the held-out test set and compared to the target variable in that set. In this study, this means the regression model is fit to 4/5 (80%) of the measurements and then predictions are made using the remaining 1/5 (20%) of measurements, and we calculated the RMSE



of those predictions. The mean RMSE across all five folds is then taken as the score for that particular hyperparameter configuration, and the set of parameters with the lowest RMSE after some predefined number of optimization trials is selected as the optimal model.

For frequency-based methods, the concept of creating predictions for a held-out set is less meaningful, because these methods use information in the input $C_{meas}$ signal across a span of times to produce their $C_{bkg}$ predictions, so holding out some data is challenging. However, to produce an RMSE score that was more comparable to that for regression methods, we produced a frequency-method $C_{bkg}$ prediction for all measurements, then calculated the RMSE for the indices associated with each of the five cross-validation folds, then took the mean of those five RMSE scores as the final score for that optimization trial. In this way, the score was a mean of scores, similar to the cross-validation approach in regression methods. We applied this same mean-of-fold's-scores approach when evaluating frequency-method predictions as in Figure 7, Figure H.1 to Figure H.3, and Table P.1. We also limited evaluation of frequency methods to use only those measurement periods where regression methods were also evaluated. We do this because the large number of predictors in regression methods gives rise to some gaps in the feature set that are not included during regression – using only those times made available to regression methods ensures a fair comparison between background stations, frequency methods, and regression methods.

We prioritized the RMSE as our regression metric due to its popularity in the literature and because it produces an error in units of the target concentration (i.e. ppmv, ppbv, or µg·m⁻³). However, we note that other metrics might produce superior model fits due to their statistical advantages. In particular, the mean squared log error (MSLE) has advantages for air pollution research, on the basis that atmospheric pollution concentrations are bounded and not normally distributed. Airborne concentrations are typically log-normally distributed, meaning a prediction error underestimating the true concentration must be bounded between zero and the true concentration, while an overestimating prediction has no upper bound. This uneven bounding means algorithms attempting to minimize the RMSE of airborne concentrations are more likely to produce a prediction that underestimates than overestimates, because the RMSE penalizes positive and negative errors equally, but only positive errors are unbounded. The MSLE, on the other hand, more strongly penalizes underestimations because it log-transforms the target and prediction, which is appropriate for air pollution concentrations where underestimations are more likely to be small due to their bounded nature. Despite these advantages, we retained the RMSE as our primary metric for the reasons mentioned above. Also, a reader can immediately understand an RMSE score in the context of typical real-world pollutant concentrations; a RMSE of 10 ppmv for a $CO_2$ prediction is understandable relative to typical real concentrations above 400 ppmv, but a MSLE of 0.001 log-ppmv is not intuitive.

The following sections describe each algorithmic $C_{bkg}$ prediction method in detail.

### Appendix C.1   Naïve rolling minimum

Baseline or background concentrations in the literature are frequently estimated as a concentration that is less than and occasionally but not always equal to the total measured concentration – in other words, the background concentration is taken to loosely follow the lower limit of measured concentrations, while transient peaks are attributed to local sources. Examples



of such approaches include those applied by Klems et al. (2010), Sabaliauskas et al. (2014), and Hilker et al. (2019). Similar approaches are also applied in other fields, such as removing baseline signals in spectroscopic signals, which share some similar characteristics to pollutant concentration signals.

Other than taking the absolute minimum measured concentration as a baseline, the next simplest approach is to consider a
rolling minimum over some period of continuous measurements. Thus a rolling minimum background has only one parameter to tune: the width of the rolling window. We considered possible window widths in the range of 5 minutes to 48 hours. Because of the simplicity of this approach, we did not apply Bayesian hyperoptimization, and instead tested all window widths in this range in 5 minute increments.

We did not expect the naïve rolling minimum model to produce reasonable estimates of background concentration. Instead,
we intended this method to serve as a bar by which to judge the remaining, more sophisticated algorithmic predictions.

### Appendix C.2    Pseudo-wavelet

The pseudo-wavelet method estimates a background concentration similarly to wavelet methods *a la* Klems et al. (2010) and Sabaliauskas et al. (2014), but it is not a true wavelet algorithm. At a high level, the pseudo-wavelet algorithm produces multiple interpolations between the two smallest values of measured downwind concentrations within a rolling window of
varying widths, and then averages these interpolations to produce a smoothed estimate of background concentration. The algorithm requires three inputs: the measured total pollutant concentrations, $C_{meas}$; the initial width of the rolling windows, $W$, in units of the $C_{meas}$ measurement frequency, which in this case was minutes; and a unitless smoothing parameter, $\alpha$.

Additional detail and applications of the pseudo-wavelet algorithm are provided by Wang et al. (2018), where it was originally introduced, and by Hilker et al. (2019), who evaluated background concentration predictions produced by the
pseudo-wavelet method against some other methods.

### Appendix C.3    Rolling ball

The rolling ball method simulates sliding a ball along the bottom of the measured total pollutant signal, with the background being the trace defined by the path of the top of the ball. This approach is common in image processing to remove uneven or noisy image backgrounds. We are not aware of any implementations of this method in air quality studies, but background
concentrations predictions from the rolling ball algorithm have similar properties to those from the pseudo-wavelet algorithm. The rolling ball method requires three inputs: $C_{meas}$, and two tuning parameters defining the shape of the ball.

In air pollution data, the horizontal axis of the $C_{meas}$ signal is in units of time while the amplitude is in units of pollution concentration. Accordingly, the rolling ball algorithm in practice is more accurately described as sliding an ellipsoid along the bottom of the $C_{meas}$ signal, with the dimensions of the ellipsoid being defined in different units from each other. The semi-
major axis of the ellipsoid will align with the concentration (vertical) axis of the pollutant signal and have units of concentration, while the semi-minor axis will align with the temporal (horizontal) axis and have units of the pollutant signal's frequency – in this case, minutes. Thus the rolling ball algorithm requires two tuning parameters which are these semi-axis lengths. To simplify this algorithm, we fixed the length of the concentration semi-axis as equal to the standard deviation of the total





measured downwind concentration, $C_{meas}$, of the relevant pollutant. Thus we reduced the number of parameters needing tuning to one. We optimized this remaining parameter, the length of the temporal semi-axis, via hyperoptimization. We considered possible widths in the range of 2 minutes to 48 hours.

**Appendix C.4    Regression model covariates**

Regression-based methods incorporated both the measured highway downwind concentration signal alongside additional predictor variables to estimate upwind background concentrations. They do not incorporate the time-series nature of the
measurements, using only concurrent values of each covariate to estimate background. We did not develop these regression models from a theoretical basis, but from a primarily statistical basis – we selected covariates for their potential to improve estimates regardless of any possible physical interpretation of their effect in a regression model.

The covariates included in each of the base regression models were:

- Total concentrations measured downwind the highway, $C_{meas}$, in units matching the pollutant.
- Concentrations measured at the two urban background stations, $C_{bkg,dv}$ and $C_{bkg,hanlans}$, also in units matching the target pollutant.
- Counts of vehicles on the highway in a given minute, $N$, in units of $veh \cdot minute^{-1}$. Only the nearest 8 of 17 lanes on the highway were captured by a radar counter.
- RLINE dispersion predictions, $k_{hwy}$, in units $s \cdot m^{-2}$.
- Squared cosine and sine of wind direction measured at the highway, $\cos^2(\theta)$, $\sin^2(\theta)$.
- Wind speed measured at the highway, $u$, in $m \cdot s^{-1}$.
- Ambient temperature measured at the highway, $T$, in °C.
- Ambient pressure measured at the highway, $P$, in hPa.
- Ambient relative humidity measured at the highway, $RH$, in %.
- Hour of day and day of week, encoded as one-hot columns for OLS and elastic net and as integers for XGBoost.

Concentrations measured at Downsview are denoted with the subscript $dv$, and Hanlan's Point with the subscript $hanlans$.

We included meteorological measurements from only the highway site, however when testing the importance of highway site-specific regression features in Section 3.3, we did not permute meteorology variables because these values tend to be strongly correlated at sites across the city and are thus not site-specific in the sense that we sought in this analysis. The purpose
of testing highway site-specific feature importance was to indirectly test model transferability, and since meteorology should be similar across sites, it does not need to be considered a site-specific feature.

For all regression models, we scaled covariates to zero mean and unit variance before fitting.

**Appendix C.5    Ordinary least squares regression**

As a first-pass regression model we employed a simple ordinary-least squares (OLS) multi-variable linear regression, with
each of the above-listed covariates as exogenous regressors. While we do not necessarily expect the relationship between



measured background concentrations and any particular covariate to be linear, we included a linear regression estimate due to the familiarity and popularity of such models in the literature.

We expect regularized and non-linear machine learning models to match or outperform OLS for all pollutants. As the naïve rolling minimum sets the bar for accuracy for all algorithmic estimates, the OLS model sets a second hurdle by which to judge
more sophisticated regression models.

### Appendix C.6     Regularized (elastic net) regression

Elastic net regression is a linear model like OLS, but applies additional penalties to model loss during fitting based on the size of regression coefficients, essentially preferring more parsimonious models with smaller coefficients. Elastic net specifically includes both L1 and L2 regularization terms, which when applied individually would be referred to as lasso and
ridge regression, respectively. The L1 penalty shrinks coefficients towards zero, penalizing large coefficients and performing variable selection. The L2 penalty shrinks large coefficients asymptotically towards zero. Applying these penalties to a linear regression model retains the interpretability of linear regression coefficients but reduces the risk of overfitting through both variable selection and coefficient shrinking. In this application, we expect the elastic net regression to outperform OLS because we test our background concentration estimates through cross-validation, which will help identify models that overfit to
training data. We selected the degrees of L1 and L2 regularization through hyperoptimization.

### Appendix C.7     Machine learning with XGBoost

Machine learning allows for non-linearity and feature interactions in the underlying relationship between true background and covariates. However, the downsides are a risk of overfitting, challenging tuning, and reduced interpretability.

XGBoost has a large number of hyperparameters to tune that can individually and together strongly influence model
performance. We selected some hyperparameters to tune and others to hold constant based on trial and error. We optimized maximum tree depth, number of boosting rounds, learning rate, L1 and L2 regularization, and XGBoost's gamma regularization term. We held other parameters constant at either their default values or at values selected through trial and error and case knowledge. We set minimum and maximum bounds for hyperparameter optimization based on best judgement and again through extensive trial and error. Selected hyperparameter values and ranges are available in the source code provided
alongside this study.

### Appendix C.8     Ensemble background estimate

As a final algorithmic $C_{bkg}$ prediction model, we considered an ensemble of predictions from each of the methods introduced thus far. Our ensemble model was an L2-regularized (ridge) regression taking each of the other estimated backgrounds (two urban background stations, three frequency methods, and three regression methods) as exogenous variables, along with an
intercept. The ensemble regression did not include the covariates listed above that were included in the base regression models, instead taking the outputs of the other models as inputs. We selected the degree of L2 regularization for the ensemble model by searching 160 logarithmically-spaced values from $10^{-9}$ to $10^{7}$, rather than through randomized Bayesian hyperoptimization.



## Appendix D    Meteorology at the highway field site

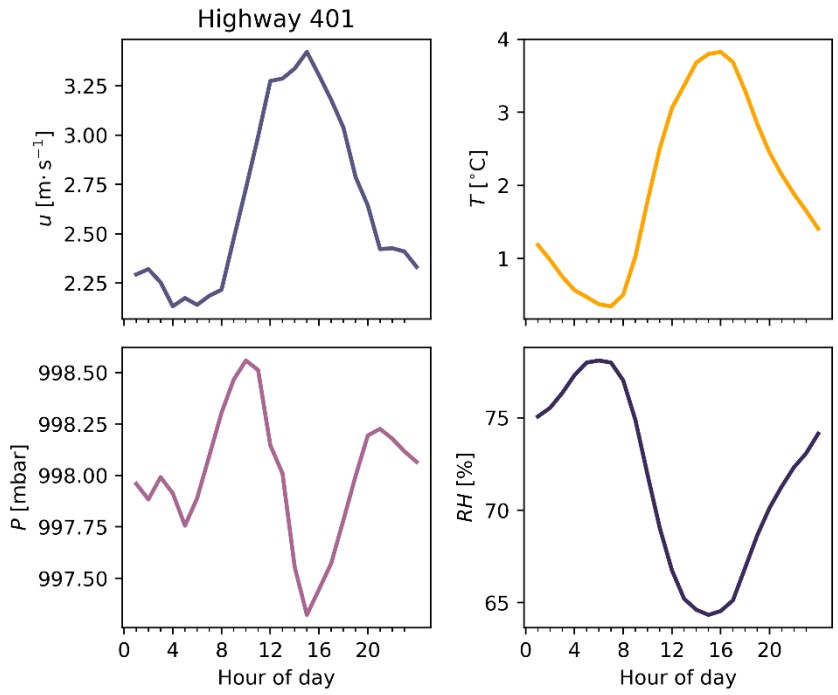


**Figure D.1. Mean diurnal patterns of wind speed ($u$), temperature ($T$), pressure ($P$), and relative humidity ($RH$) measured at the Highway 401 downwind south station.**

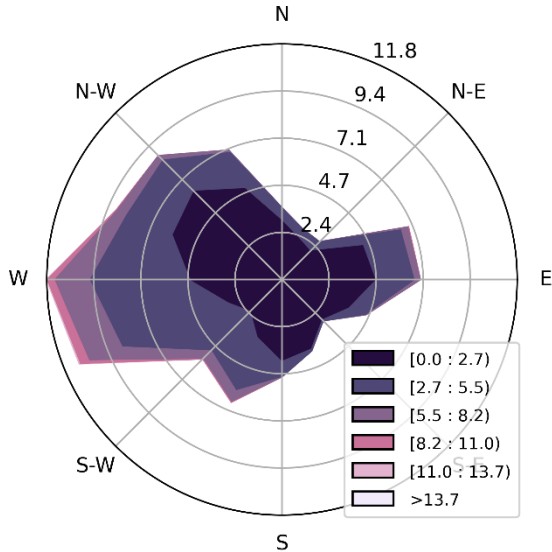

**Figure D.2. Wind rose depicting dominant wind speeds and directions at the Highway 401 field study location, measured on the**
**south and predominantly downwind side of the highway.**



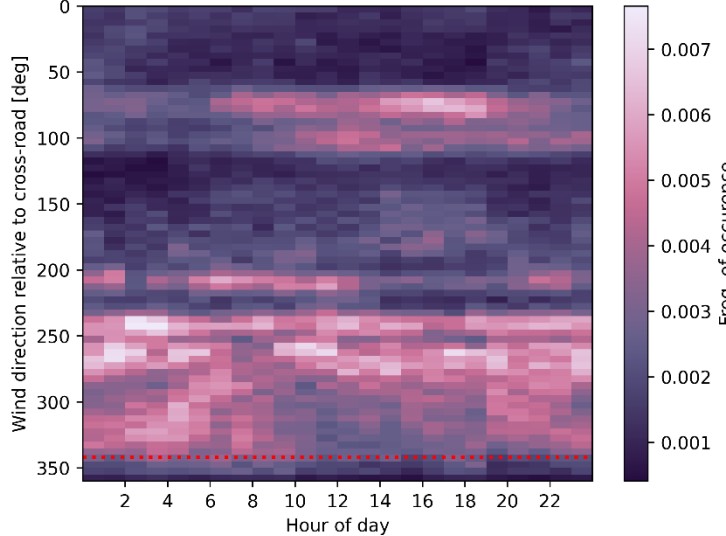

**Figure D.3. Diurnal heatmap depicting frequency of wind directions measured on the south side of Highway 401 over the entire study period. The red dashed line indicates the direction that would be directly perpendicular and across the road at the measurement point.**






## Appendix E    Comparing SHARP and T640 instruments



**Figure E.1: Scatter matrix comparing SHARP and T640 instruments across three of the sites used in this study. Red lines are one-to-one, dashed lines with shaded areas are linear regressions with 95% confidence intervals.**





## Appendix F      Separating local and background signals by wind speed and direction

Figure F.1 shows background and roadside downwind concentrations at Highway 401 as a function of concurrent wind direction. From this figure, we identified the wind directions appropriate for considering the background sensor north of the highway to be a true measure of $C_{bkg}$. As indicated in the methodology, the range we selected was between 80 degrees to the northwest and 40 degrees to the northeast, based largely upon the ranges where the difference in down- and upwind sensors (i.e. $C_{local} = C_{meas} - C_{bkg}$) began to trend towards zero.

In addition to decreasing mean concentrations concurrent with the higher wind speeds as discussed in the methodology and visible in Figure F.2, we also observed an unexpected maximum mean $C_{local}$ for some pollutants at wind speeds ~2 m · s$^{-1}$. This was most apparent for NO$_x$, but was also present to a lesser extent in CO and CO$_2$. The cause of increasing $C_{local}$ at wind speeds below 2 m · s$^{-1}$ is not clear. With all other variables (meteorology, emissions, etc) held constant, simple dispersion theory predicts decreasing local concentrations associated with increasing wind speeds. There are some possible explanations for this observation: higher wind speeds typically occur during midday to afternoon when insolation is greatest, which is concurrent with higher anthropogenic activity and thus emissions. This possibility is supported qualitatively by Figure F.3, which shows similar trends of $C_{local}$ as a function of wind speed but with the underlying measurements coloured by time of day also shown. In these figures, we observed that higher wind speeds and higher concentrations both tended to occur later in the day – more green points are to the right of the axes in Figure F.3, indicating that we recorded higher wind speeds more often later in the day. These simultaneous correlations lend themselves to the appearance of a positive correlation between wind speed and $C_{local}$. This can be corroborated by comparing the diurnal trends of $C_{local}$ in Figure I.1 and wind in Figure D.1, where we observed high average concentrations during the same times of day as high average wind speeds.



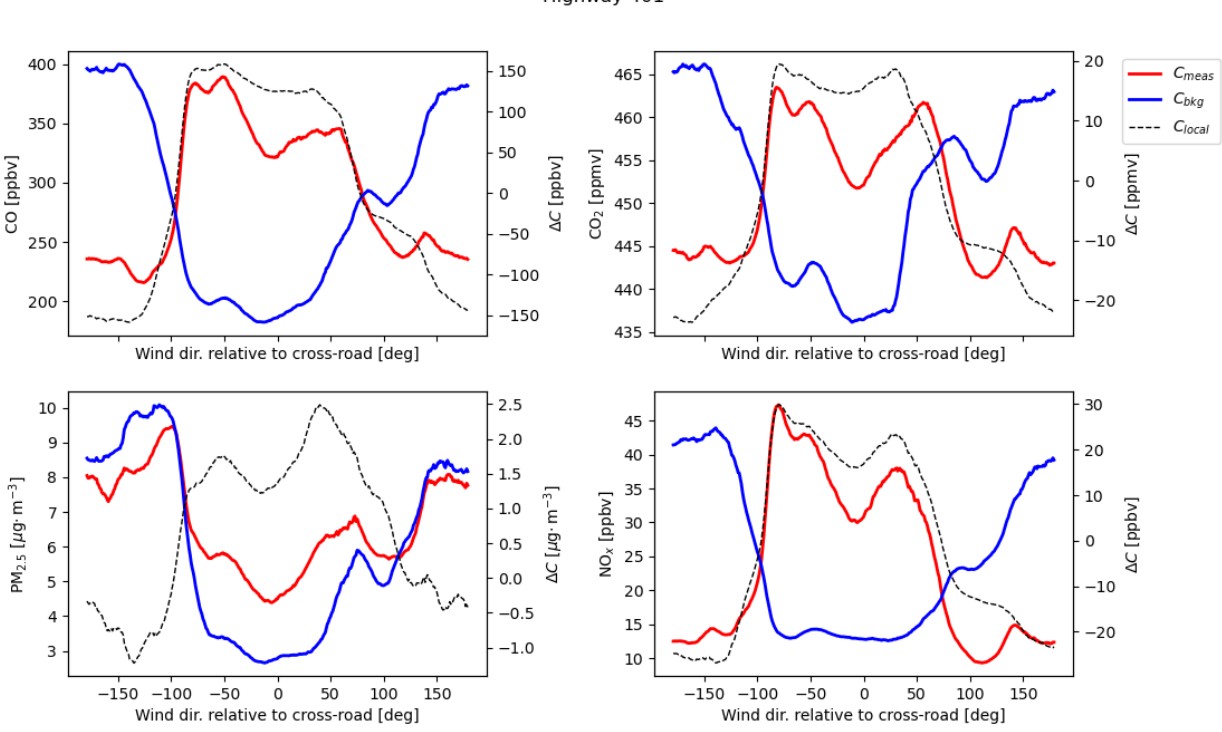

**Figure F.1. Mean pollutant concentrations at the Highway 401 site binned by concurrent wind direction in one degree bins. Wind direction is adjusted so zero is directly normal and facing across the road from the roadside downwind measurement site. The highway lays mostly east-west, so positive directions indicate more easterly winds and negative directions indicate more westerly winds. Trends were smoothed and interpolated with a weighted centred rolling mean across 15 adjacent increments, weighted by sample size.**

945




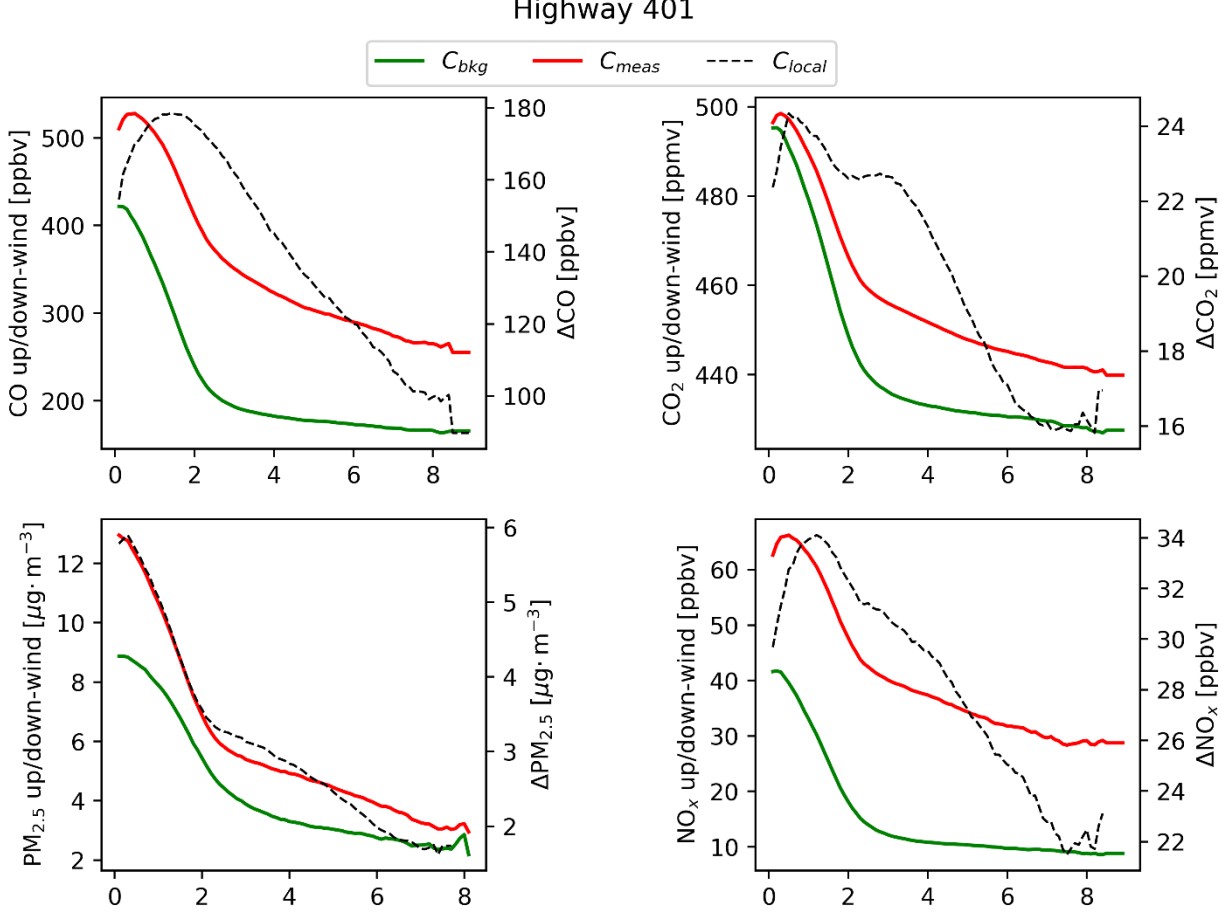

**Figure F.2. Mean pollutant concentrations at the Highway 401 site as a function of concurrent wind speed. Trends were generated by first calculating mean concentrations within $0.25 \text{ m} \cdot \text{s}^{-1}$ bins of concurrent wind speeds. Increments with fewer than 60 measurements were excluded. Trends were smoothed and interpolated with a weighted centered rolling mean across 11 adjacent increments, weighted by sample size. For $C_{local}$, we only included periods were $C_{local} > 0$ when producing these trends.**



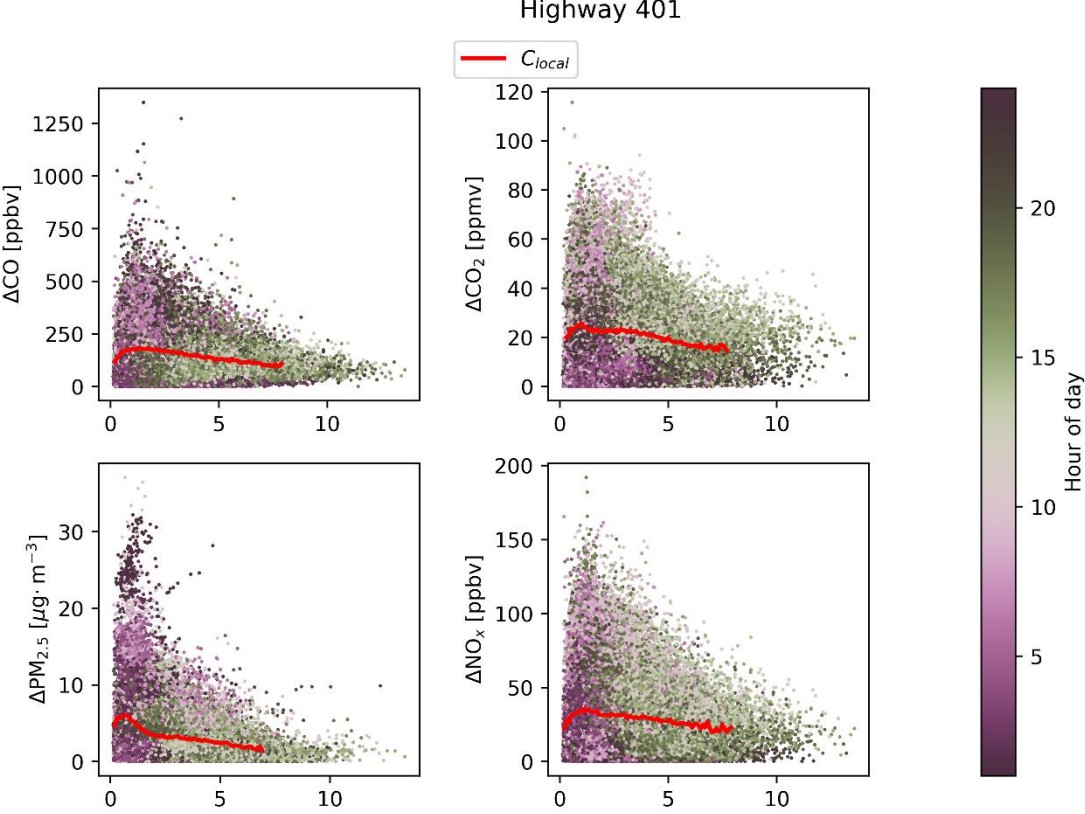

**Figure F.3. Mean pollutant concentrations at the Highway 401 site as a function of concurrent wind speed. Mean trends were generated by taking mean concentrations within 0.25 m · s$^{-1}$ bins of concurrent wind speeds. Points are underlying measurements used to generate the trends and are coloured by hour of day the measurement fell within. We only included periods where $C_{local} >$ 0 when producing these scatters and trends.**



## Appendix G        Exemplar time-series trends



**Figure G.1. Example of measured and estimated background pollutant signals at the Highway 401 field study site. For clarity, not all background estimation methods are shown here. Grey shaded regions indicate when the south site was downwind the highway, indicating periods where the $C_{bkg}$ signal was a valid measurement of background concentration as defined in the methodology.**



## Appendix H $C_{bkg}$ prediction accuracies for $NO_x$, $CO_2$, and $PM_{2.5}$

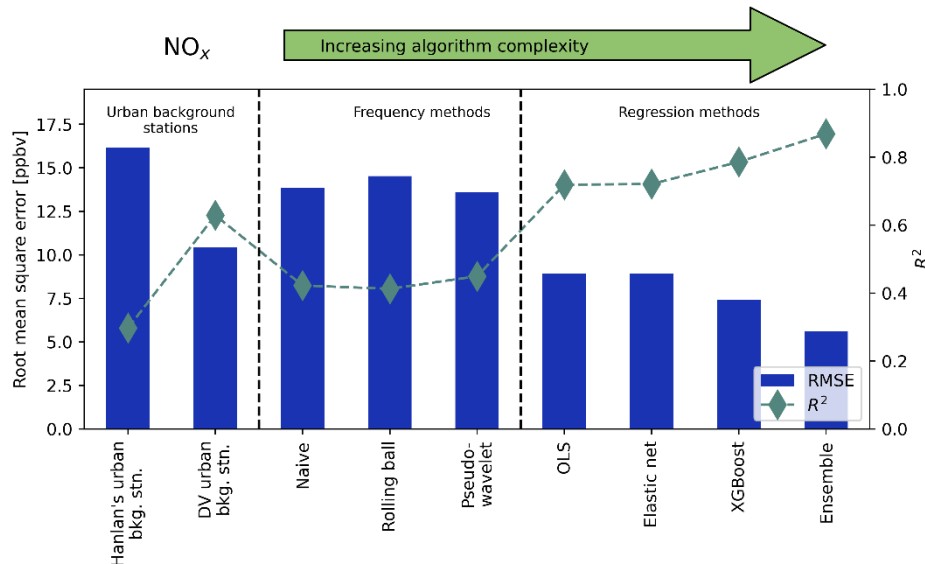

**Figure H.1. Root mean square error (RMSE, bars) and coefficient of determination ($R^2$, diamonds) for predicted background $NO_x$ at the highway site, as predicted by each method tested here. Scores show the accuracy of each method in estimating true upwind background concentration, with lower RMSE and greater $R^2$ being better. Scores were calculated as the mean across five-fold cross-validation.**

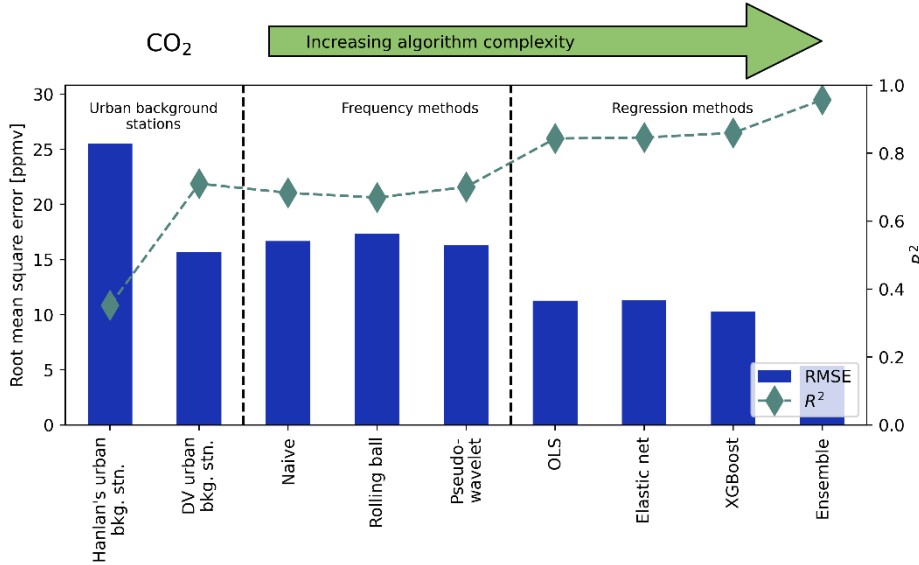

**Figure H.2. Root mean square error (RMSE, bars) and coefficient of determination ($R^2$, diamonds) for predicted background $CO_2$ at the highway site, as predicted by each method tested here. Scores show the accuracy of each method in estimating true upwind background concentration, with lower RMSE and greater $R^2$ being better. Scores were calculated as the mean across five-fold cross-validation.**



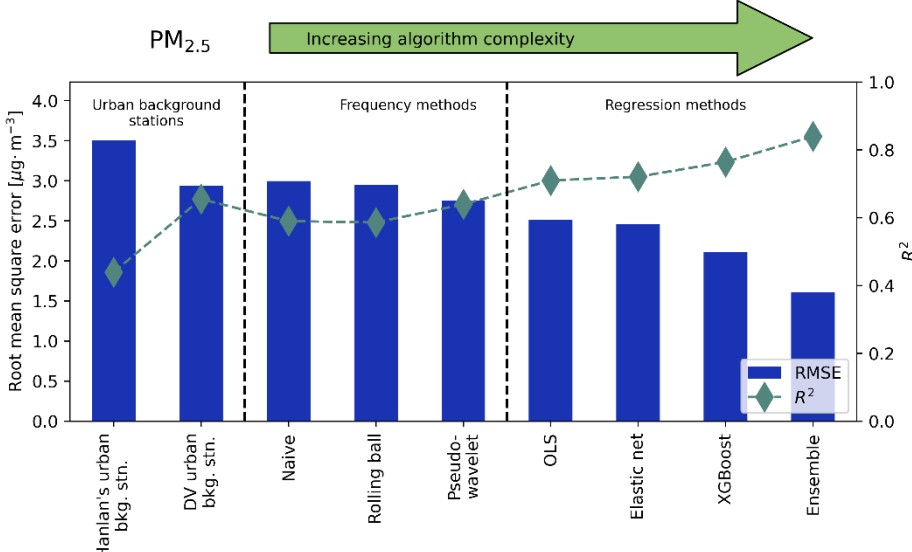

**Figure H.3. Root mean square error (RMSE, bars) and coefficient of determination ($R^2$, diamonds) for predicted background PM2.5 at the highway site, as predicted by each method tested here. Scores show the accuracy of each method in estimating true upwind background concentration, with lower RMSE and greater $R^2$ being better. Scores were calculated as the mean across five-fold cross-validation.**





**Appendix I** $C_{local}$ **diurnal patterns**

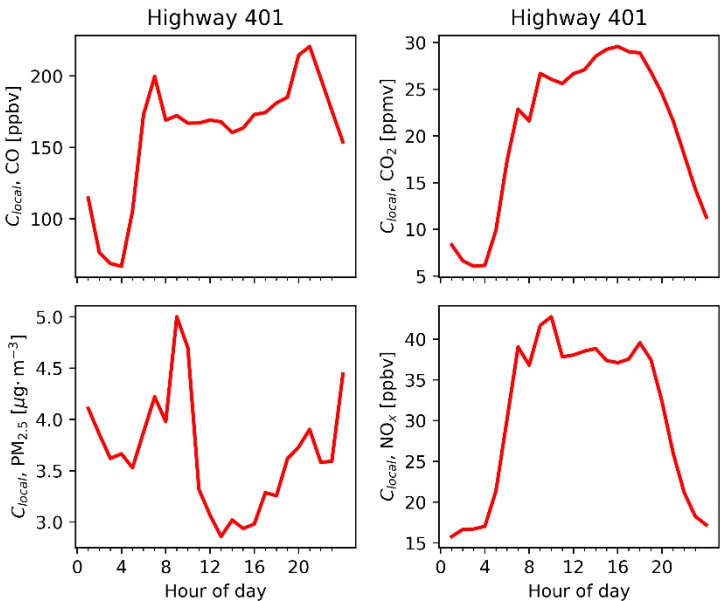

**Figure I.1. Mean hourly diurnal trends of the difference between measured concentrations downwind the highway ($C_{meas}$) and background concentrations upwind of the highway ($C_{bkg}$) for each pollutant. Periods where the difference, $C_{local}$, was negative, were excluded.**





 **Appendix J**      **Background concentration scatters**



**Figure J.1. Paired scatters and kernel density estimates (KDE) of background carbon monoxide concentrations at three stationary measurement sites in the Greater Toronto Area. Red lines are 1-to-1. For the Highway 401 site, backgrounds were only considered valid when wind direction and speed fell within the ranges specified in the methodology; figures only show periods where**
**backgrounds were concurrently measured at each site. To speed calculation of the KDE and lessen figure density, a random 20% subset of measurements are shown here.**





**Figure J.2. Paired scatters and kernel density estimates (KDE) of background nitrogen oxides (NO + NO₂) concentrations at three stationary measurement sites in the Greater Toronto Area. Red lines are 1-to-1. For the Highway 401 site, backgrounds were only considered valid when wind direction and speed fell within the ranges specified in the methodology; figures only show periods where backgrounds were concurrently measured at each site. To speed calculation of the KDE and lessen figure density, a random 20% subset of measurements are shown here.**







**Figure J.3. Paired scatters and kernel density estimates (KDE) of background particulate matter < 2.5 μm diameter concentrations**
**at three stationary measurement sites in the Greater Toronto Area. Red lines are 1-to-1. For the Highway 401 site, backgrounds**
**were only considered valid when wind direction and speed fell within the ranges specified in the methodology; figures only show**
**periods where backgrounds were concurrently measured at each site. To speed calculation of the KDE and lessen figure density, a**
**random 20% subset of measurements are shown here. Note that the Hanlan's Point site used a different PM₂.₅ instrument – see**
**methodology for details.**





**Appendix K        Frequency method optimized hyperparameters**

While frequency methods were often less accurate in predicting $C_{bkg}$ than regression methods, they can provide insight into background pollutant behaviour by examining their optimized hyperparameters. For the naïve rolling minimum and rolling ball algorithms, both were fit with a single hyperparameter, and in both cases this single parameter expresses an effective width

of temporal duration of measured roadside downwind concentrations to consider when estimating background concentrations. For the naïve rolling minimum, the tuned parameter is the window width in minutes, and for the rolling ball axis it is the radius along the temporal semi-axis of the ellipse that is "rolled" along the bottom of the downwind pollution concentration signal. For both, a larger parameter produces a predicted $C_{bkg}$ that has less or slower temporal variability, and a lower average magnitude. For the pseudo-wavelet method there are two parameters that are somewhat interchangeable in how they affect the

resulting $C_{bkg}$ prediction, but they can be similarly interpreted because larger values again produce more slowly-varying and smaller signals.

Table K.1 shows the hyperoptimized best parameters for each frequency method. The differences between optimized parameters reflected the characteristics and spatial variability of the pollutants – in particular, the order of pollutants as ranked by frequency method coefficients loosely correlated with pollutants as ordered by their coefficients of variation (CV) in Table

2. $NO_x$ and $PM_{2.5}$ had the largest hyperparameters across methods and the greatest CVs, followed by CO, and then $CO_2$. Another way to interpret these parameters is to consider that for all frequency methods, very large hyperparameters lead to background predictions that approach a constant value, so the relative size of these parameters indicates the extent to which the background concentration for that pollutant might be appropriately estimated as a constant value. Thus these parameters provide additional, albeit indirect, evidence for differences in temporal variability of pollutant backgrounds relative to their

means. This correlates with our prior observation that low $NO_x$ background concentrations paradoxically make predicting $NO_x$ $C_{bkg}$ both easier and harder depending on the context.

**Table K.1. Hyperoptimized parameters for the naïve rolling minimum, rolling ball, and pseudo-wavelet (PW) background estimation algorithms. Parameters are in units of minutes except $\alpha$, which is unitless.**

|            | Naïve | Ball | PW $\alpha$ | PW $W$ |
|------------|-------|------|-------------|--------|
| CO         | 115   | 185  | 15          | 16     |
| $CO_2$     | 45    | 86   | 7           | 19     |
| $NO_x$     | 210   | 289  | 23          | 23     |
| $PM_{2.5}$ | 175   | 360  | 22          | 22     |

For the pseudo-wavelet algorithm, the ranking of optimal $\alpha$ and $W$ parameters were similar to the naïve minimum and rolling ball methods. Larger values of $W$ produce background concentration predictions that vary more slowly and less frequently equal the input $C_{meas}$ signal, and thus make up a smaller portion of the total measured concentration. In other words, larger values of $W$ indicate that local emissions are a more dominant driver of concentration variability. Similar conclusions



can be drawn for values of $\alpha$. However, to a certain extent $W$ and $\alpha$ are interchangeable, as demonstrated by the examples in
Hilker et al. (2019), so it is more challenging to draw meaningful conclusions about background concentration characteristics
from the pseudo-wavelet algorithm's parameters than from the naïve and rolling ball methods, which each use a single and
more easily interpreted tuning parameter. Despite this, we find a broad agreement across frequency methods in the relative
magnitudes of optimized parameters between pollutants: these parameters suggest $NO_x$ and $PM_{2.5}$ background concentrations
varied less rapidly relative to their average levels than $CO_2$ and CO.





**Appendix L** **Elastic net regression coefficients**

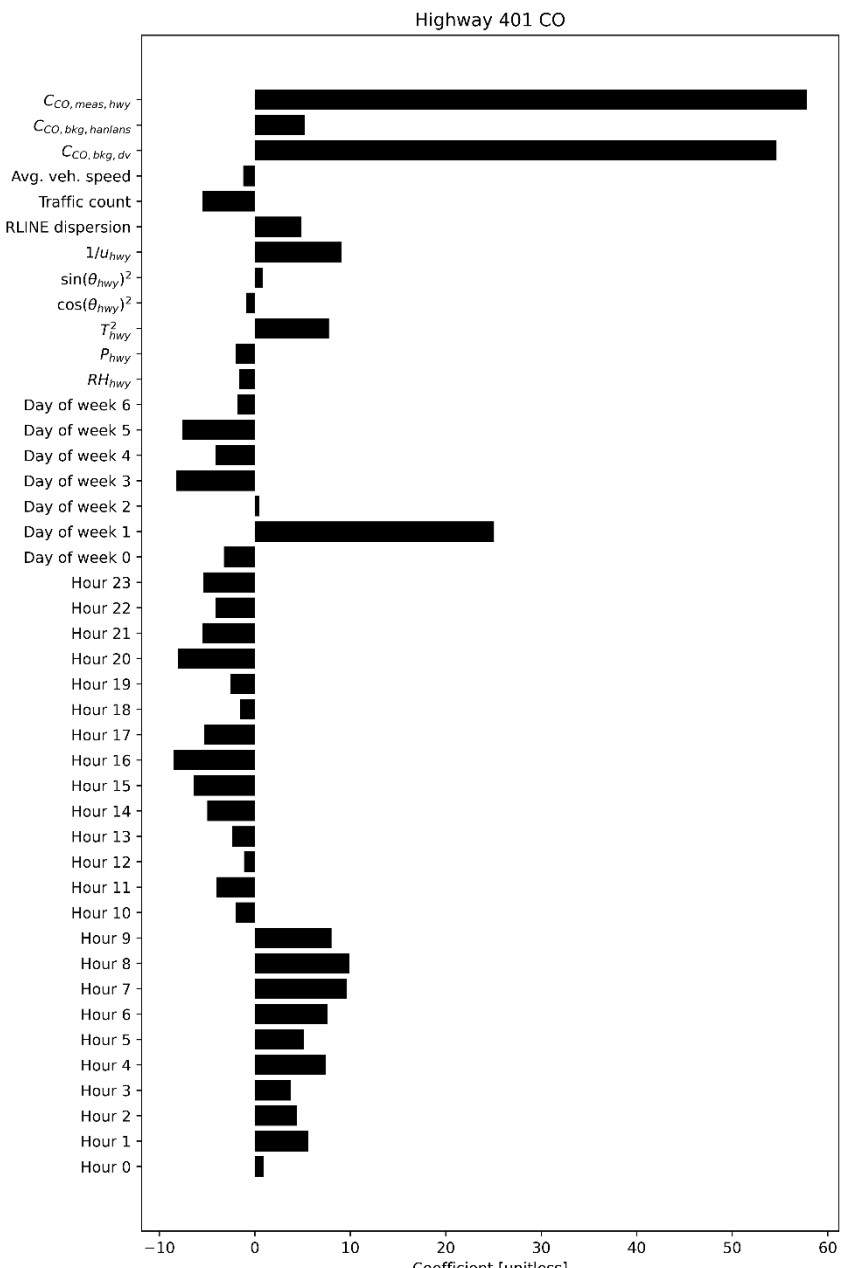

**Figure L.1. Elastic net regression coefficients for predicted highway upwind background CO. The optimal degree of L1 and L2 regularization was identified via five-fold stratified cross-validation. Covariates and target concentrations were standardized prior to fitting, so coefficients are unitless.**



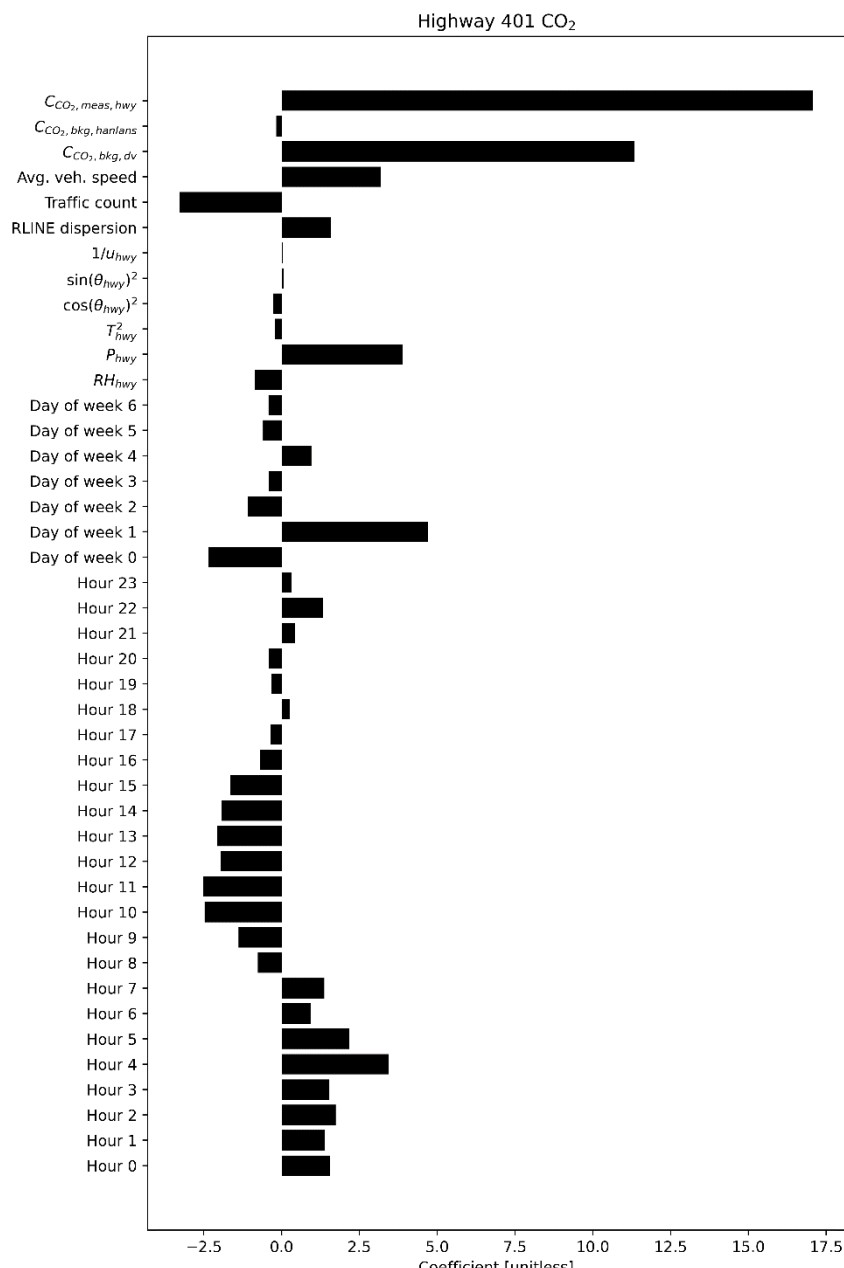


**Figure L.2. Elastic net regression coefficients for predicted highway upwind background CO₂. The optimal degree of L1 and L2 regularization was identified via five-fold stratified cross-validation. Covariates and target concentrations were standardized prior to fitting, so coefficients are unitless.**



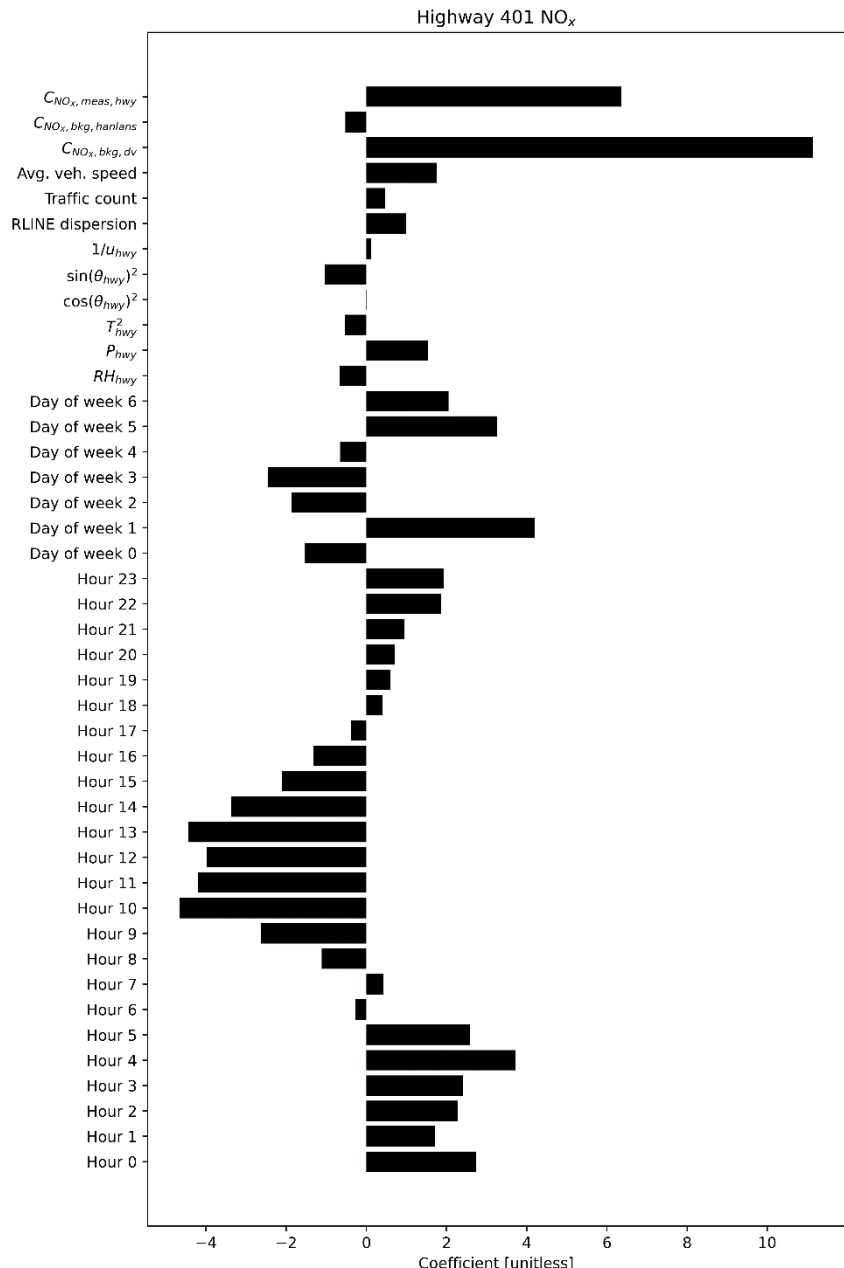

**Figure L.3. Elastic net regression coefficients for predicted highway upwind background NO$_x$. The optimal degree of L1 and L2 regularization was identified via five-fold stratified cross-validation. Covariates and target concentrations were standardized prior to fitting, so coefficients are unitless.**



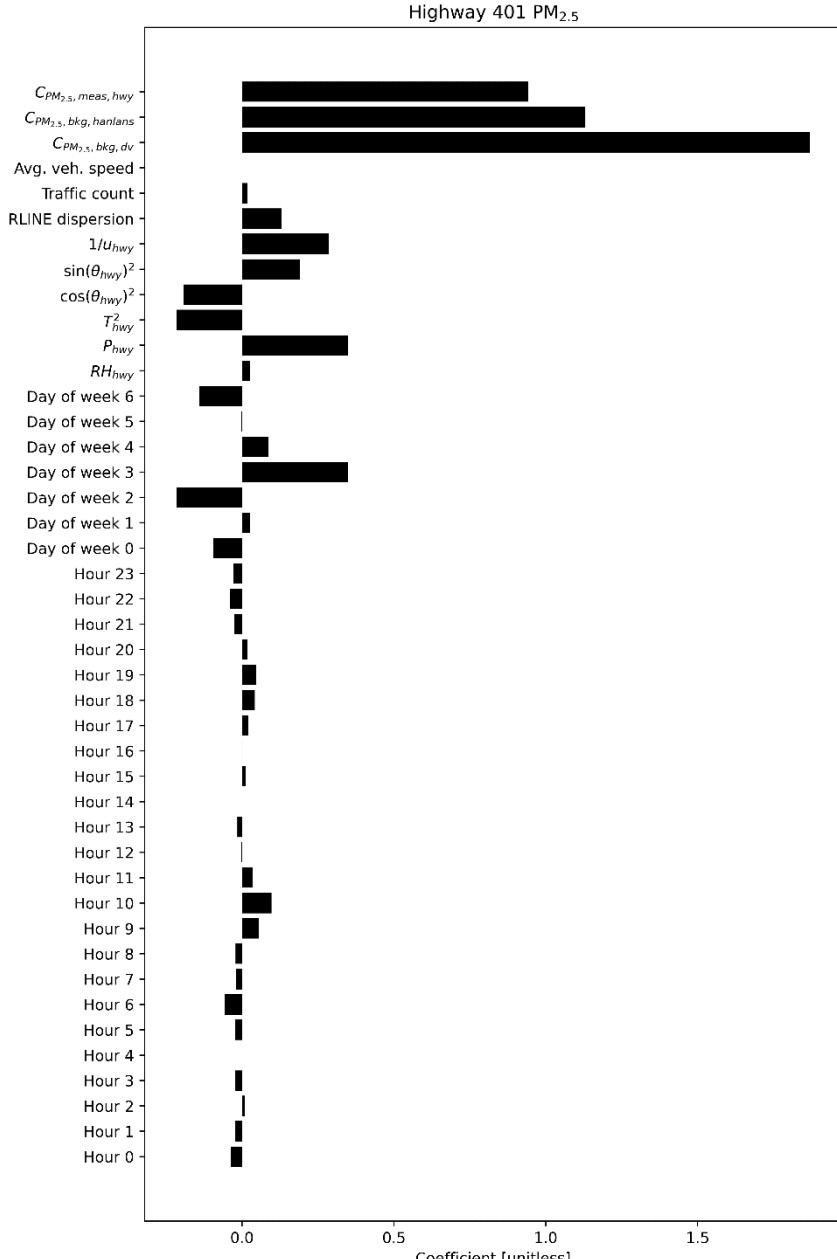

**Figure L.4. Elastic net regression coefficients for predicted highway upwind background PM$_{2.5}$. The optimal degree of L1 and L2 regularization was identified via five-fold stratified cross-validation. Covariates and target concentrations were standardized prior to fitting, so coefficients are unitless.**





## Appendix M    Example of cross-validation stratification



**Figure M.1. Train-test split with five-fold cross-validation for predicting highway upwind background NO$_x$. This example figure demonstrates how measurements were split during cross-validation. In each fold, models were trained on measurements coloured blue and tested against measurements coloured orange. Black points demonstrate model-predicted background concentrations in each fold.**





## Appendix N    S



**Figure N.1. Measured background pollutant concentrations at the highway site as a function of each covariate considered in regression models. To clarify potential underlying relationships, covariates were binned into 50 equally spaced bins based on their individual ranges; dots are means of the background concentrations within that bin, and bars are standard deviations. Only periods with valid measures of $C_{bkg}$, as defined in the methodology, are included.**





**Appendix O        Ensemble ridge regression trace plots**

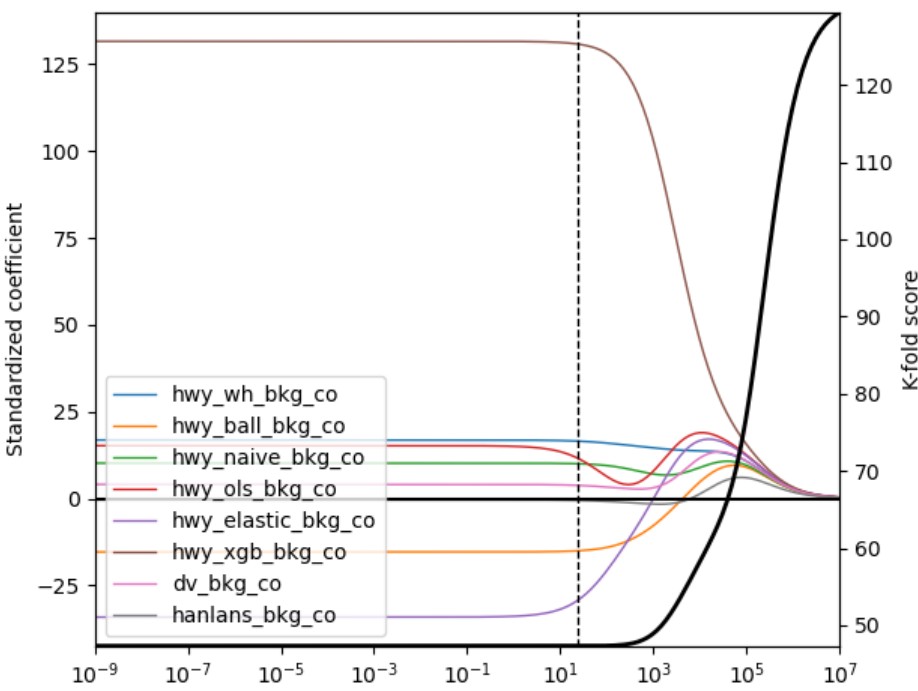


**Figure O.1. Highway upwind background CO ensemble model ridge regression coefficients and mean five-fold cross-validation score as a function of regularization strength. Coloured traces are regression coefficients and the thick black trace is root mean squared error of predicted concentrations. The dashed vertical line indicates the degree of regularization with the lowest error.**



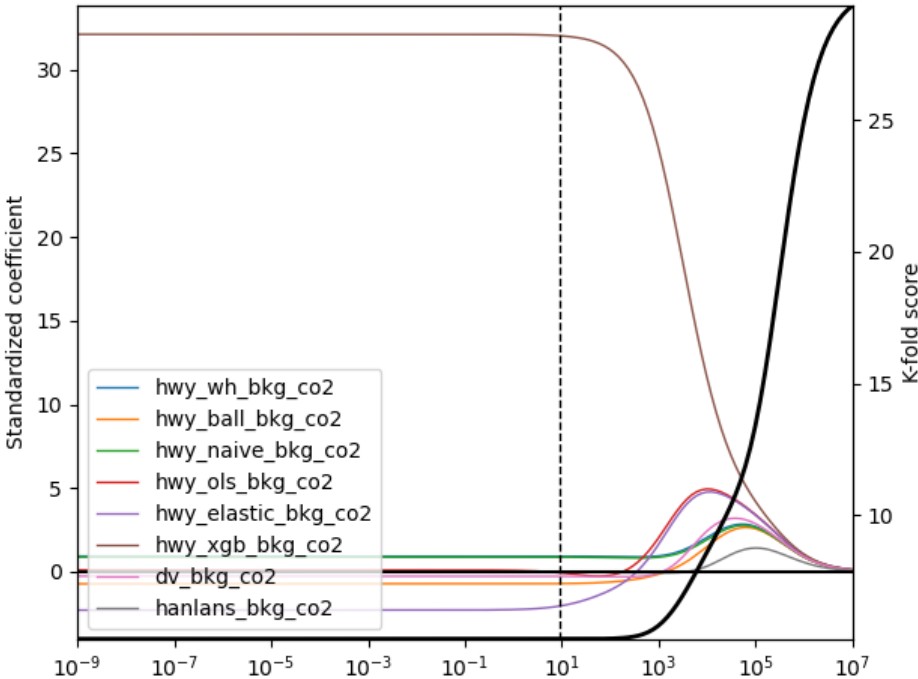

**Figure O.2. Highway upwind background CO₂ ensemble model ridge regression coefficients and mean five-fold cross-validation score as a function of regularization strength. Coloured traces are regression coefficients and the thick black trace is root mean squared error of predicted concentrations. The dashed vertical line indicates the degree of regularization with the error.**





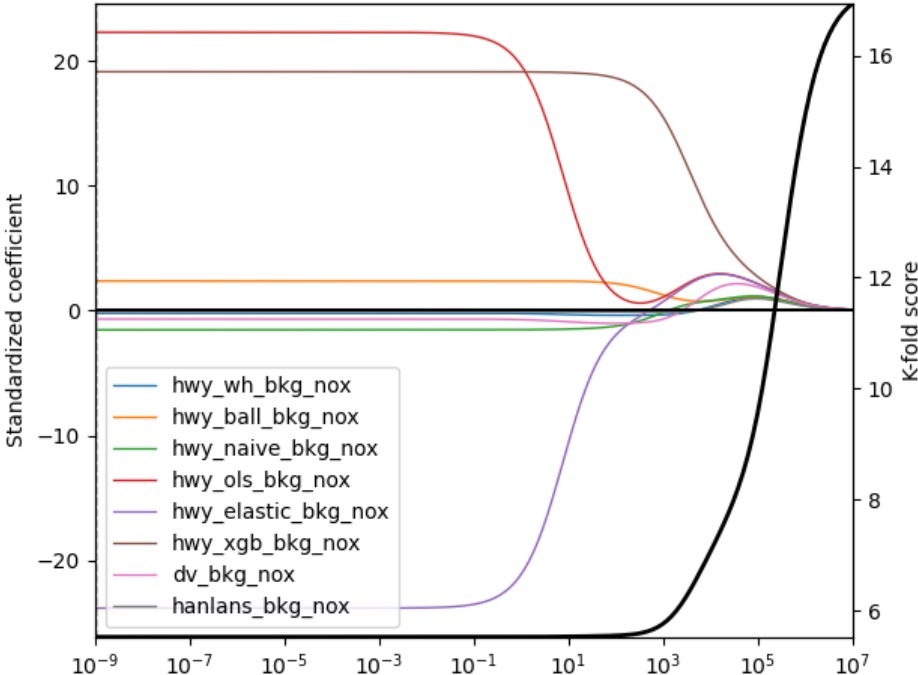

**Figure O.3. Highway upwind background NO$_x$ ensemble model ridge regression coefficients and mean five-fold cross-validation score as a function of regularization strength. Coloured traces are regression coefficients and the thick black trace is root mean squared error of predicted concentrations. The dashed vertical line indicates the degree of regularization with the lowest error.**



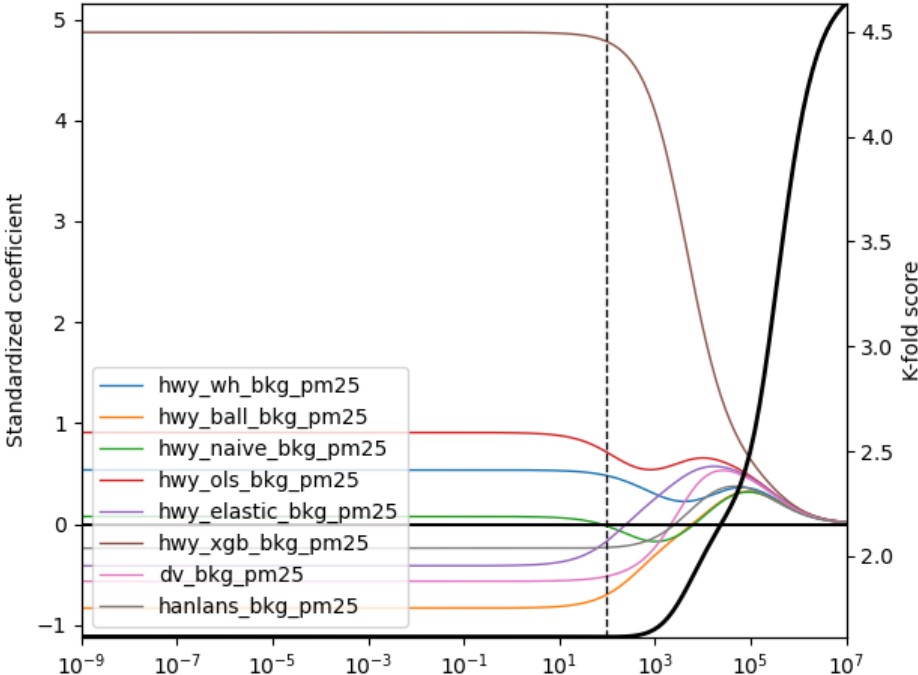

**Figure O.4. Highway upwind background PM$_{2.5}$ ensemble model ridge regression coefficients and mean five-fold cross-validation score as a function of regularization strength. Coloured traces are regression coefficients and the thick black trace is root mean squared error of predicted concentrations. The dashed vertical line indicates the degree of regularization with the lowest error.**




## Appendix P    Tables of background concentration estimate accuracy metrics

**Table P.1. Performance statistics of each background prediction method when compared to true measured $C_{bkg}$ at the highway upwind background, as defined in the methodology. Scores are the means across folds in five-fold cross-validation. Values were rounded to three significant figures.**

| Pollutant | Prediction method | RMSE | $R^2$ | Spearman's R | $F_{1.1}$ | $m_g$ | FB | Max error | Mean error | Min error | MAE |
|---|---|---|---|---|---|---|---|---|---|---|---|
| | Downsview | 89.8 | 0.59 | 0.742 | 0.352 | 0.893 | 0.163 | 1280 | 36.6 | -639 | 48.7 |
| | Hanlan's Point | 110 | 0.375 | 0.63 | 0.4 | 0.996 | 0.11 | 1320 | 25.8 | -510 | 55.8 |
| | Naïve rolling min. | 92 | 0.561 | 0.644 | 0.198 | 1.21 | -0.116 | 1220 | -29.6 | -223 | 65.5 |
| | Pseudo-wavelet | 89.5 | 0.633 | 0.704 | 0.161 | 1.27 | -0.157 | 1200 | -40.9 | -179 | 68.6 |
| CO | Rolling ball | 96.8 | 0.542 | 0.659 | 0.182 | 1.25 | -0.14 | 1190 | -36.1 | -217 | 70.3 |
| | OLS | 72.2 | 0.665 | 0.728 | 0.328 | 1.02 | -0.00066 | 1180 | -0.223 | -487 | 46.3 |
| | Elastic Net | 71.9 | 0.678 | 0.759 | 0.345 | 1.02 | -0.00184 | 1190 | -0.401 | -466 | 44.8 |
| | XGBoost | 63.5 | 0.727 | 0.757 | 0.469 | 1.03 | -0.0136 | 1160 | -3.03 | -332 | 37.6 |
| | Ensemble | 47.6 | 0.824 | 0.88 | 0.636 | 1.01 | 0.00037 | 1120 | 0.13 | -294 | 25.2 |
| | Downsview | 15.7 | 0.71 | 0.655 | 0.972 | 0.997 | 0.00768 | 103 | 3.41 | -69.4 | 9.77 |
| | Hanlan's Point | 25.5 | 0.352 | 0.638 | 0.941 | 0.995 | 0.0158 | 161 | 7 | -55.5 | 13.4 |
| | Naïve rolling min. | 16.7 | 0.683 | 0.558 | 0.981 | 1.01 | -0.0123 | 100 | -5.5 | -51 | 12.5 |
| | Pseudo-wavelet | 16.3 | 0.701 | 0.565 | 0.984 | 1.02 | -0.0123 | 84.7 | -5.52 | -49.6 | 12.4 |
| $CO_2$ | Rolling ball | 17.4 | 0.669 | 0.558 | 0.978 | 1.02 | -0.0136 | 100 | -6.11 | -51.6 | 13 |
| | OLS | 11.3 | 0.843 | 0.792 | 0.994 | 1 | -0.00022 | 74.3 | -0.107 | -51.6 | 7.96 |
| | Elastic Net | 11.3 | 0.846 | 0.798 | 0.994 | 1 | -0.00021 | 75.7 | -0.106 | -52 | 7.9 |
| | XGBoost | 10.3 | 0.86 | 0.84 | 0.995 | 1 | -0.00197 | 74 | -0.876 | -56.7 | 6.96 |
| | Ensemble | 5.34 | 0.957 | 0.942 | 1 | 1 | 4.92E-06 | 58.6 | 0.00366 | -39.3 | 3.56 |
| | Downsview | 2.93 | 0.655 | 0.678 | 0.102 | 0.649 | 0.192 | 29.8 | 0.93 | -38.3 | 2.06 |
| | Hanlan's Point | 3.5 | 0.439 | 0.696 | 0.184 | 1.07 | -0.0816 | 31.2 | -0.386 | -30.8 | 1.9 |
| | Naïve rolling min. | 3 | 0.59 | 0.643 | 0.118 | 0.913 | 0.0793 | 30.3 | 0.551 | -13.4 | 1.95 |
| | Pseudo-wavelet | 2.75 | 0.639 | 0.668 | 0.143 | 0.976 | 0.0668 | 30.1 | 0.499 | -8.27 | 1.77 |
| $NO_x$ | Rolling ball | 2.95 | 0.586 | 0.645 | 0.124 | 0.99 | 0.0473 | 30.2 | 0.424 | -9.8 | 1.92 |
| | OLS | 2.51 | 0.71 | 0.798 | 0.168 | 1.09 | -0.0483 | 29.8 | -0.0396 | -18 | 1.58 |
| | Elastic Net | 2.46 | 0.721 | 0.801 | 0.179 | 1.11 | -0.0485 | 29.8 | -0.0309 | -13.9 | 1.54 |
| | XGBoost | 2.11 | 0.764 | 0.847 | 0.27 | 1.06 | -0.0202 | 29.9 | 0.051 | -7.71 | 1.22 |
| | Ensemble | 1.61 | 0.84 | 0.896 | 0.354 | 1.03 | -0.0108 | 29.4 | 0.03 | -7.05 | 0.904 |
| | Downsview | 10.4 | 0.629 | 0.551 | 0.11 | 0.713 | 0.253 | 95.6 | 3.49 | -76.7 | 6.8 |
| | Hanlan's Point | 16.2 | 0.297 | 0.392 | 0.0494 | 0.532 | 0.556 | 131 | 6.97 | -60.5 | 9.5 |
| | Naïve rolling min. | 13.9 | 0.422 | 0.389 | 0.1 | 1.11 | -0.116 | 126 | -2.05 | -46.3 | 9.25 |
| | Pseudo-wavelet | 13.6 | 0.449 | 0.401 | 0.107 | 1.22 | -0.158 | 111 | -2.89 | -39.3 | 9.23 |
| $PM_{2.5}$ | Rolling ball | 14.5 | 0.413 | 0.422 | 0.104 | 1.33 | -0.236 | 120 | -4.42 | -45.9 | 9.95 |
| | OLS | 8.91 | 0.719 | 0.693 | 0.148 | 1.04 | -0.00075 | 86 | -0.249 | -56.7 | 6.01 |
| | Elastic Net | 8.92 | 0.721 | 0.694 | 0.145 | 1.04 | 0.000101 | 86.2 | -0.252 | -56.8 | 6 |
| | XGBoost | 7.4 | 0.786 | 0.742 | 0.206 | 1.06 | -0.0311 | 81.2 | -0.594 | -58.6 | 4.66 |
| | Ensemble | 5.59 | 0.869 | 0.823 | 0.264 | 1.01 | 0.00207 | 75.5 | -0.0057 | -45.6 | 3.51 |




*Code and data availability.* Analysis code is available in a Zenodo repository with DOI: 10.5281/zenodo.13236885. Raw data may be made available upon request.


*Author contributions.* CRediT: Taylor D. Edwards: conceptualization, methodology, software, validation, formal analysis, resources, data curation, writing – original draft, writing – review & editing, visualization. Yee Ka Wong: methodology, data collection validation, investigation, resources, writing – review & editing. Jonathan M. Wang: investigation, resources, data curation. Cheol-Heon Jeong: investigation, data collection resources, data curation. Yushan Su: data collection, review and

editing. Greg J. Evans: writing – review & editing, supervision, project administration, funding acquisition.

*Competing interests.* The AirSENCE air quality monitoring technology was originally developed at the Southern Ontario Centre for Atmospheric Aerosol Research at the University of Toronto and it has now been commercialised and is being distributed by AUG signals, with licensing fees paid to the University of Toronto.


*Funding:* The Natural Science and Engineering Research Council and Environment and Climate Change Canada provided funding to support this research.

*Acknowledgements.* We would like to acknowledge everyone, past and present, involved in maintaining the three

stations operated by Ontario Ministry of the Environment, Conservation and Parks and the Wallberg laboratory operated by the Southern Ontario Centre for Atmospheric Aerosol Research at the University of Toronto.



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
