# Peer review of "Comparison of methods for resolving the contributions of local emissions to measured concentrations"

_EGUsphere, 2024_

## Referee Comment (RC1)

**Review – *Resolving the contribution of local emissions to measured concentrations: a method comparison**

In this paper, the authors used measurements of nitrogen oxides (NOx), carbon monoxide (CO), carbon dioxide (CO2), and particulate matters (PM2.5) at four sites in Toronto, Ontario, Canada, to characterize the spatial variability of background concentration across the city and compare several prediction background concentration algorithms with increasing complexity. The focus is on the prediction of background concentration close to a busy Highway (Highway 401). They show that the prediction quality increases with complexity of the algorithms and further provide a ranked recommendation, based on several aspects.

This is an interesting and important topic for determining local sources. A clear separation of background concentration from local or nearby sources is an important task, especially if there is no designated measuring location for the respective case or is not possible.
This is significant, not only for pollutant mentioned here, but for observation sites in general measuring long- and short-lived substances with potential anthropogenic or natural near sources as well. The work is very comprehensive and detailed, writing and illustrations are in general good, but not everywhere clear.
Structurally, I would suggest some improvements to make it easier for the reader to follow the statements and connections in this work with some more explanations in certain passages and a revision of the extensive appendix.

I would therefore recommend this manuscript after major revisions.

**Major comments:**

**Title revision**
The papers focus is the evaluation of the seven different algorithmic methods to estimate the true background concentration near the Toronto's Highway 401, although the overall goal is to do this to estimate local concentrations. For now, the title does not yet give a good indication of what to expect in this paper. I suggest a revision of the title, for example *"Comparison of background concentration prediction algorithms to determine the contribution of local emissions to the measured concentration"* or similar, to pinpoint to the scope of the paper.

**Structure of introduction and methodology**
The authors give very nice and details review of notable background-subtraction methods used in the field of air quality studies. Further, a clear outline of the objectives within this publication is given.
I would suggest reconsidering the subsection 1.1 and 1.2 to make the introduction clearer. The definition of background concentration (Subsection 1.1) in the context of this work would also be well suited as the first subsection in the methods as it is a definition which differs from existing interpretations of background in air pollution research. In addition, the situation of the study is already discussed here (line 101-102). Subsection 1.2 can then be included in Section 1 (no subsection anymore). For this, I would suggest deleting the part from line 115 to 126 but include at the end of the introduction briefly how the paper is organized.

Subsection 2.2 "*Separating measured local and background concentration at the highway*" already discussed results of the measurement site at the Highways 401. Thus, I am not sure how well this fits to the methodical part of this paper or maybe be part of the results. Further two figures from the Appendix are discussed within this section and should be part of the main text and not the appendix (see next comment).

***Extend of the appendix***

This manuscript is followed with a very long appendix with many references to figures and discussions within the appendix (e.g., line 259, 284, and 480). For my knowledge, a research paper must be complete without appendices and must contain all information including tables, diagrams, and results necessary to address the research problem. For this, I would suggest some changes to the appendix by include some of the information into the main text as well as excluding figures, which are not mentioned in the main text (at least I could not find mentions) or include mentions, where needed to support the result.

My suggestion for inclusion in the main text:
- Appendix F, with selected figures from Figure F1 and F2, as these are important for comparison with results from the algorithms.

My suggestion for general excluding from appendix:
- Figure D.2 is already included in Figure 1 and thus redundant.
- Figures D.1, D.3, E.1, G.1, I1, M.1, N.1, O.1, O.2, O3., and O.4 are not mentioned explicitly in the main text and should be revisited.
- Table P.1 is not mentioned in the main text as well.

**Minor comments:**

Line 18        Why explicit mention XGBoost? May comment on that.

Line 21        "outperformed" sounds not appropriate (only personal opinion). How about "surpassed the performance of …"

Line 23        Here and in the following: the usage of the semi-colon is new to me. As a non-native speaker, I am not familiar with the use of the semi-colon as seen here in the work. From the sentences I could read here, you could also start a new sentence instead of using the semi-colon. This makes it a little easier for the reader to follow.

Line 52        Would change "wholly" by "completely".

Line 58        As a reader who is not completely familiar with the topic, I am not sure what "signal decomposition …" means. This should be clarified.

Line 65        It may be nice to provide a brief description of the wavelet decomposition method here or later in the text.

Line 85-114    It is very important that the authors note, that the here mentioned "background" concentration is different to the regular definition of background concentration, usually associated with baseline or clear-air

concentration. This is done partly in line 108-110. Rather, in the context of the publication, background concentration is meant to be concentration from sources other than very close local sources. I think, this section can be shortened and can be part of the methods section, as this is self-chosen definition of background concentration in the context of this work with the example of the configuration at the highway field.

Line 135       "outperform", see comment line 21.

Line 154       As I get it right, you looked at four and not five sites throughout Toronto. Further, I would recommend change the style of the period from *"2023-11-23 to 2024-04-12"* to *"from 23 November 2023 to 12 April 2024".*

Line 156       This section could be included into 2.1 Field measurements. Thus, Section 2.1 would not be only one sentence.

Line 157-158  Here and in the following, where the locations of the measurements are described.  I am not familiar with the UTM coordination system and other readers might as well are not familiar with this system. I would suggest using geographical coordinate system (GCS). Also, include *"(see A in Figure 1, top; Figure 1 bottom)"* at the end of this sentence.

Line 163-165  Reference to the bottom Figure could be included where necessary.

Line 170       The two figures could also be displayed in the same width.

Line 175       Here and in the following, add the letters as in figure 1 (top).

Line 189-194  Something went wrong with the font size.

Line 209       Zheng et al. is missing the year.

Line 222       What is the reason for averaging to the nearest minute? Please give some more details.

Line 235       Figure F.1: These are very interesting figures. The y-axis title should be more precise or should be explained in more detail.

Line 237       The information of the wind direction to isolate local and background signals is very important. I find it a little bit difficult to read the wind direction *"between 80 degrees to the northwest and 40 degrees to the northeast"* in combination with the x-axis in figure F1, which is *"wind dir. relative to cross-rad [deg]"*. I would recommend a uniform wind direction for the text and the figures, e.g., simple degree without relative to the road.

Line 240       Figure F.2: These are very interesting figures. Please provide titles for the x-axis for clarity

Line 247-259 This paragraph is about periods, where $C_{meas}$ could be smaller than $C_{bkg}$. As the last sentence already stated, that additional discussion can be found in the appendix, I would suggest shortening this paragraph, e.g. *"When applying measurements or estimates of background concentrations, in some applications it would be useful to further limit valid measurements of $C_{bkg}$ to periods where $C_{meas} \geq C_{bkg}$. In our analysis here we chose not to remove periods where $C_{meas} < C_{bkg}$ to avoid eliminating too great a portion of our measurements from our analysis, and to acknowledge that for pollutants where background concentration makes up a large portion of the whole measured concentration (as is the case for CO2 and PM2.5), the difference between $C_{meas}$ and $C_{bkg}$ can be small enough that instrument sensitivity will play a role in determining if the difference between the two is measurable. Further discussion regarding this topic can be found in Appendix F."*

Line 266-268 I would suggest not listing the stations again as these are described before.

Line 327-342 I think the discussion about the low NOx background concentrations are a little bit too long and could be shortened a little bit.

Line 345 Why are the two first boxplots in the same color? I suggest using different color for all.

Line 419 Please also give the full name of KDE in the text, not just in the figure caption.

Line 443-445 If this part is important for the manuscript, it should be included into the main text. I would not recommend refer to a discussion in the appendix.

Line 454 Figure 5 is mentioned in the text but is not discussed further. I would suggest a few more lines about what to see in the figure, especially discussing the superimposed lines of Highway upwind bkg. and XGboost bkg.

Line 459 Where are the solid red lines in the plots as mentioned in the caption?

Line 475 A description of the 1:1 line is missing here or in the text. Maybe it is worth to calculate the orthogonal distance regression line to compare which prediction fits the best.

Line 481 I would leave out words like "unsurprisingly" or use something like "As already shown in Figure 6 ..."

Line 490 I would not use words like 'interestingly' in a scientific paper. The work should be objective to be robust and credible without a personal option, what in the text is more interestingly than others.

Line 545-548 I am not sure, how the indirect conclusion was made ("... together contain most of the information necessary..."). Maybe, because I am not too familiar with this topic. It would be great to get more information about how this conclusion was made.

Line 549      Make again clear what is meant with "This lack of difference …"

Line 577      See comment to line 490 but for "surprisingly".

Line 647-648  The ensemble model is a bit lost in the work, although it shows the best
              results. It is mentioned again in lines 687-688 and I understand that, although
              it shows the best results, it requires the output of all other methods and thus
              the effort is the greatest. Nevertheless, it should perhaps be reconsidered
              what part the ensemble model should play in this work.

**Reviewed by an Early Career Scientist**

---

## Author Response (AR1)

**Author's response for the article titled *Comparison of methods for resolving the contributions of local emissions to measured concentrations**

We thank the two anonymous reviewers for their comments. We have reviewed each reviewer's comments and edited our paper accordingly. Our point-by-point replies to these comments are listed in our submitted responses to each reviewer comment.

In addition to the edits made in response to reviewer comments, we also made minor changes throughout to fix typos, improve grammar, increase clarity, etc. We also updated fig. 9, though the discussion of this figure was largely unchanged.